# Mapping risks of health conditions in people with atopic eczema in English primary care and hospital data

Julian Matthewman [1] ✉, Anna Schultze[1], Krishnan Bhaskaran[1], Amanda Roberts[2], Spiros Denaxas[3,4,5], Kathryn E. Mansfield [6] & Sinéad M. Langan [1]

Atopic eczema may be associated with multiple health conditions. Here, we systematically explore risks across the full health spectrum based on the International Classification of Diseases, assessing associations between eczema and 2058 ICD-10 codes, 1593 phecodes, and 201 Global Burden of Disease codes. In English primary care electronic health records (1997 – 2023) we identify cohorts of people with eczema (up to 3 million) and matched (by age, sex, general practice) comparators without eczema (up to 14 million). In up to 25 years of follow-up, we capture outcomes recorded during hospital admissions. People with eczema show higher rates of several outcomes across multiple organ systems. Among those followed up from eczema diagnosis in childhood, atopic/allergic conditions and infections account for most excess diagnoses. Consistent across cohorts and analyses, large relative risk increases are seen for inflammatory bowel conditions (e.g., K50 Crohn disease, crude hazard ratio 1.70 [1.63-1.77]) and eye diseases (e.g., H16 Keratitis, crude hazard ratio 1.71 [1.57-1.86]). We provide a dashboard to explore and browse the full range of results.

Eczema, also referred to as atopic eczema or atopic dermatitis, is common and associated with the subsequent occurrence and co-occurrence of other atopic diseases such as asthma and allergies. Associations with subsequent non-atopic diseases are less clear. There is no internationally accepted approach to screening for, or prevention of, associated outcomes. It is possible that this lack of a consensus approach to prevention is due to clinicians not knowing which conditions to prioritise[1,2].

A previous study by our team offered evidence on a range of outcomes – recorded in primary care – chosen for their inclusion in guidelines or previous research[1,3]. Knowledge of risks across the full health spectrum faced by people with eczema is, however, still limited, and there may be associated conditions that had not previously been considered as associated with eczema. Furthermore, ascertainment bias may be an issue when assessing outcomes in primary care, as individuals with eczema may consult more frequently due to their eczema.

Diagnoses in UK hospital records are structured using the International Classification of Diseases (ICD) Version 10; a hierarchical terminology of which the 2058 codes of the third level ("category-level") correspond to diseases (e.g., asthma, atopic dermatitis, scabies, impetigo, psoriasis, etc.), symptoms, or health-related occurrences. ICD-10 codes can also be mapped to other disease classification systems, including phecodes (covering 1593 diseases) to provide more detail on certain conditions[4], and Global Burden of Disease codes (covering 201 diseases) to provide a higher-level overview of disease burden[5].

[1]London School of Hygiene & Tropical Medicine, London, United Kingdom. [2]Independent Patient Partner, London, United Kingdom. [3]Institute of Health Informatics, University College London, London, United Kingdom. [4]NIHR UCLH BRC, London, United Kingdom. [5]BHF Data Science Centre, London, United Kingdom. [6]School of Health and Care Sciences, University of Lincoln, Lincoln, United Kingdom. ✉e-mail: julian.matthewman@lshtm.ac.uk

Here, we explored associations between eczema and outcomes recorded as part of hospital admissions. The hierarchical structure of ICD-10 allowed us to systematically learn about risks across the full health spectrum. Our aim was to create a comprehensive evidence base that facilitates comparison between outcomes, prioritisation in clinical practice, and reduces the risk of ascertainment bias. We used multiple disease classification systems (ICD-10, phecodes, Global Burden of Disease codes) as each has different strengths and limitations.

## Results

### Descriptive statistics

From English primary care data from the Clinical Practice Research Datalink (CPRD) Aurum linked to Hospital Episode Statistics (HES) Admitted Patient Care (APC) data from April 1st 1997 to March 31st 2023 ($N = 35,968,445$), we identified 3,513,875 individuals who met the eczema definition ($>1$ record for eczema and $>2$ records for eczema therapies in primary care) and were eligible for matching. We matched them with up to five eligible individuals who had no record of eczema at the diagnosis date of the exposed people. Without restrictions on age (any-age cohort) this resulted in a cohort of 17,180,124 (with and without eczema); 7,478,587 when including only individuals who met the eczema definition before their 18th birthday (<18 cohort), 12,772,961 for a cohort of individuals at least 18 years of age (18 + cohort), and 7,075,624 for a cohort of individuals at least 40 years of age (40 + cohort) (Fig. 1). For analyses of each specific outcome, individuals who already had a hospital record for that outcome of interest at index date were excluded (e.g., when assessing the association with asthma, 1,545,003 [9%] who already had a hospital record for asthma were excluded; for stroke 113,085 [0.7%] were excluded). We assessed the presence of primary-care-recorded comorbidities at index date, for some of which there were considerable differences between groups (e.g., previous asthma 12% in unexposed versus 21% in exposed) (Supplementary Table 1).

For each outcome, individuals were followed up using their hospital admission records until they had the outcome of interest (event) recorded in any diagnostic position (i.e., primary or secondary reasons for admission or medical history), or were censored. Median follow-up time was 5.5 (interquartile range 2.1, 11.0) years in the any-age cohort, 4.8 (1.9, 9.9) for the 18 + cohort, 6.2 (2.7, 11.3) for the 40+ cohort, and 5.2 (2.0, 10.8) for the <18 cohort (in the unexposed 4.7 [1.8, 10.0], 4.4 [1.8, 9.3], 5.8 [2.5, 10.8], and 3.9 [1.5, 9.1], respectively) (Supplementary Table 1).

### Associations between eczema and subsequent hospital-recorded diagnoses

For each outcome of interest, we provide hazard ratios and rate differences from Cox regression from crude and comorbidity-adjusted models, by age cohort (any-age, < 18, 18 +, 40 +), and from sensitivity analyses. In Table 1 we show examples of synthesising all available information for selected outcomes. The full range of results can be found either in Supplementary Materials or using the Interactive Figures and Tables in the dashboard available at https://github.com/julianmatthewman/Eczema_hospital_outcomes_public.

### Statistical significance

Of the 2058 tested associations between eczema and outcomes based on ICD-10 codes (crude results from Cox regression), 44% (any-age), 11% (<18), 44% (18 +), and 40% (40 +) had p-values below the Bonferroni-corrected 1% significance level (0.01/2058 = 0.000005) (Supplementary Table 2, **Interactive** Fig. 2). The percentage of significant outcomes decreased to 36% (any-age), 8% (<18), 30% (18 +), and 29% (40 +) when adjusting for comorbidities at baseline, and 21% (any-age), 2% (<18), 17% (18 +), and 17% (40 +) when also excluding non-consulters (individuals who likely did not see their GP in the year prior to index date).

### Excess rate

Some of the largest excess rates of recorded outcomes in people with eczema were seen in Chapters R (clinical signs and symptoms) (e.g., rate difference per 1000 person-years [RD] for Abnormalities of breathing: 1.22 in any-age cohort; 1.66 in the < 18 cohort), and Z (health status) (e.g., RD for Personal history of risk factors: 1.34 in any-age cohort; 1.48 in < 18 cohort). Together, outcomes in chapters R and Z made up more than a quarter of the total excess burden (calculated as the sum of the excess rate in people with eczema for each outcome) in both the any-age and < 18 cohorts. L30 Other dermatitis had a large excess rate in all cohorts (3.29 in any-age cohort; 3.96 in < 18 cohort) (Fig. 2, **Interactive** Fig. 1).

When excluding chapters L (skin), Q (congenital conditions), and R-Z (symptoms and other health-related occurrences), diagnoses in Chapters J (respiratory) and A-B (infections) accounted for more than half of the total excess burden in the < 18 cohort. In the any-age cohort, the total excess disease burden was more evenly distributed across chapters. The largest rate differences in the any-age cohort were for J45 Asthma (4.41), I10 Essential hypertension (2.05), F32 Depressive episode (1.05), and J18 Pneumonia, organism unspecified (0.97). For the < 18 cohort, it was J45 Asthma (4.13), B34 Viral infection of unspecified site (1.36), J30 Vasomotor and allergic rhinitis (0.60), J35 Chronic diseases of tonsils and adenoids (0.53), and J22 Unspecified acute lower respiratory infection (0.51) (Fig. 3 and Supplementary Table 2).

**Hazard ratios.** Excluding Chapters L and Q-Z and outcomes with < 1000 events in the eczema group, in the any-age cohort hazard ratios were largest for J46 Status asthmaticus (2.95 [2.79–3.13]), B00 Herpesviral [herpes simplex] infections (2.52 [2.38–2.67]), J30 Vasomotor and allergic rhinitis (2.14 [2.08–2.20]), J45 Asthma (2.12 [2.11–2.14]), M32 Systemic lupus erythematosus (1.79 [1.65–1.95]), K90 Intestinal malabsorption (1.78 [1.72–1.85]), M07 Psoriatic and enteropathic arthropathies (1.75 [1.64–1.87]), J31 Chronic rhinitis, nasopharyngitis and pharyngitis (1.75 [1.66–1.86]), B35 Dermatophytosis (1.72 [1.58–1.86]), and H16 Keratitis (1.71 [1.57–1.86]). In Fig. 3 we show hazard ratios together with the excess rate. In Figs. 4, 5 we show hazard ratios and excess rate for the 5 outcomes with the largest crude HRs in each chapter. Crude and adjusted HRs for all outcomes are shown in Supplementary Table 2, 3 and **Interactive** Table 1.

Hazard ratios from adjusted models are shown in Supplementary Table 2-3. There were substantial differences between crude and adjusted hazard ratios for some outcomes (e.g., asthma: crude HR 1.78 [1.74-1.82], adjusted HR 1.51 [1.46-1.56], change − 15.12%) but not for others (e.g., essential [primary] hypertension: crude HR 1.14 [1.12–1.16], adjusted HR 1.14 [1.12–1.16], change −0.06%)

### Sensitivity analyses

In Supplementary Table 3 we provide results from all age cohorts. In Supplementary Tables 3, 4 we provide results from sensitivity analyses (the any-age cohort including only individuals that had been hospitalised at least one year before the index date, and from all cohorts excluding non-consulters). In both sensitivity analyses, hazard ratios were generally attenuated.

### Mappings to phecodes and GBDcodes

In Figs. 6, 7 and Supplementary Table 5, and **Interactive** Table 2 we show results for outcomes mapped to 201 Global Burden of Disease (GBD) codes, which are ordered hierarchically (e.g., A. Malignant neoplasms: HR 1.09 [1.08–1.10], 108,133 events; 22. Lymphomas, multiple myeloma: HR 1.29 [1.15-1.34], 7749 events; and a. Hodgkin lymphoma: HR 1.71 [1.54−1.90], 969 events). In Figs. 8, 9, Supplementary

**(a) Study design diagram**

**(b) Example timeline**

**(c) Study flow diagram**

**Fig. 1 | Study diagrams. a** Study design diagram, (**b**) example timeline, and (**c**) Study flow diagram, colour-coded by step. [a] Treatments include emollients, topical glucocorticoids, topical calcineurin inhibitors, systemic immunosuppressants (azathioprine, methotrexate, ciclosporin, mycophenolate), and oral glucocorticoids. [b] Unexposed individuals are censored on the day they meet the eczema diagnostic algorithm themselves, and can then be re-matched, this time as exposed individuals. [c] Example outcomes are J45 "Asthma" and I63 "Cerebral infarction".

Table 6**, and Interactive** Table 3 we show results mapped to 1593 phecodes, which have a different hierarchical structure (e.g., 201 Hodgkin's disease: HR 1.71 [1.54-1.9], 969 events; 202 Cancer of other lymphoid, histiocytic tissue: HR 1.33 [1.26-1.4], 3,821 events; 202.2 Non-Hodgkins lymphoma: HR 1.32 [1.27-1.38], 4824 events; 202.21 Nodular lymphoma: HR 1.16 [1.04-1.29], 790 events; 202.24 Large cell lymphoma: HR 1.31 [1.22-1.42], 1,690 events). In Supplementary Table 7 we show the 6 strongest associations per phecode category.

## Discussion

Here, we produced a comprehensive atlas of results on how eczema is associated with other diseases, assessing the full health spectrum. Our results both give an overview of excess risk across all outcomes, as well as provide a range of results from different cohorts and analyses for each individual outcome (we demonstrate how the full range of results can be used in Table 1).

**Table 1 | Example associations**

| Disease | Relative strength of association (Crude hazard ratios from any-age cohort; see Figs. 3, 4 and Supplementary Table 2, 3) | Absolute strength of association (Crude rate differences from any-age cohort; see Figs. 2, 4 and Supplementary Table 2) | Adjusted results (Primary care recorded comorbidity adjusted results; see Supplementary Table 1 for recorded comorbidities; see Supplementary Table 2, 3) | Results by age (Results from <18, 18+, and 40+ cohorts; <18 cohort: exposed individuals met the eczema definition before their 18th birthday; 18+ / 40+ cohorts: individuals at least 18/40 years; see Supplementary Tables 2, 3) | Results from sensitivity analyses (Results from ever hospitalised cohort and excluding non-consulters; see Supplementary Tables 3, 4) | Related outcomes and primary-care results (if available) see Supplementary Tables 2, 4 for outcomes, and Supplementary Table 10 for results from previous primary-care study) | Phecodes (see Supplementary Table 6 for phecode-mapped outcomes) | Summary |
|---|---|---|---|---|---|---|---|---|
| J45 Asthma | Crude HR of 2.12 [2.11–2.14]. Within 2% of the strongest relative associations (rank 31 of 1419 outcomes with > 100 events). Strongest (Status asthmaticus) and fourth strongest (asthma) associations in the Diseases of the respiratory system (J00-J99) chapter. | Largest crude RD of any outcome of 4.4/ 1000 person-years. Largest crude RD in the <18 cohort of 4.1/ 1000 person-years. Next largest RDs are for L30 Other dermatitis (3.29), and Z86 PH of certain other diseases (2.30). | Strong attenuations of effect for both J46 Status asthmaticus (from 2.95 [2.79–3.13] to 2.29 [2.15–2.44]; a change of −22.56%), and J45 Asthma (to 1.61 [1.59–1.62]; a change of −24.20%). Asthma in primary care available in adjustment set. | HR decreases with cohort age: 2.50 [2.46–2.53] (<18), 2.12 [2.11–2.14] (any-age), 1.98 [1.96–2.00] (18 +), 1.84 [1.82–1.87] (40 + cohort). Adjusted results (respectively): 2.06 [2.02–2.09], 1.61 [1.59–1.62], 1.34 [1.33–1.36], 1.31 [1.29–1.33]. | Attenuated in ever hospitalised cohort to 1.93 [1.91–1.95] (adjusted to 1.51 [1.50–1.53]). Larger attenuation of crude HR to 1.72 [1.69–1.74] when excluding non-consulters. | J46 Status asthmaticus with larger crude HR (any-age): 2.95 [2.79–3.13]. Generally larger HR in all other analyses, but fewer events recorded than J45 asthma. R06 Abnormalities of breathing has the 4th largest RD in the <18 cohort. Previous primary-care results for asthma were similar (adjusted HR of 1.87 for asthma in any-age cohort). | Phecode 495 "asthma" covers J45 asthma. Phecode 495.2 "Asthma with exacerbation" covers J46 Status asthmaticus. | Strong relative and one of the strongest absolute strengths of association especially in childhood. Results match those seen with primary care recorded asthma. Effect estimates are attenuated in adjusted analyses (including adjustment for asthma recorded in primary care at baseline). |
| K50 Crohn disease [regional enteritis] | Crude HR of 1.70 [1.63-1.77]. Within 4% strongest relative associations (Rank 59 of 1419 outcomes with >100 events). | Crude RD of 0.12/ 1,000 person-years. (rank 200 of 1419 outcomes with >100 events). Similar crude rate differences for K51 Ulcerative colitis (both 0.12). | Attenuation to 1.47 [1.41-1.54]; a change of 13.42%. Crohn's disease in primary care is available in adjustment set. | HR decreases with age of cohort: 1.80 [1.65-1.96] (<18), 1.70 [1.63-1.77] (any-age), 1.65 [1.58-1.72] (18 +), 1.56 [1.47-1.64] (40 +). Adjusted results (respectively): 1.67 [1.53-1.82], 1.47 [1.41-1.54], 1.35 [1.29-1.42], 1.29 [1.22-1.36]. | Attenuated in ever hospitalised cohort to 1.55 [1.48-1.62] (adjusted to 1.38 [1.32-1.45]). Similarly large attenuation of crude HR to 1.44 [1.35-1.52] when excluding non-consulters. | Association with K51 Ulcerative colitis (also an inflammatory bowel disease) are weaker but still strong (crude HR of 1.51 [1.46–1.56]). Previous primary-care results for Crohn's disease were similar (adjusted HR of 1.62 in any-age cohort). | Phecode 555.1 "Regional enteritis" covers K50 Crohn disease [regional enteritis]. Phecodes 555.2 "Ulcerative colitis" and 555.21 "Ulcerative colitis (chronic)" show crude HRs of 1.51 [1.46-1.56] and 1.62 [1.52-1.73], respectively. | Strong relative but small absolute association. Results match those seen in primary care. Associations stronger than for ulcerative colitis. |
| I10 Hypertension | Crude HR of 1.24 [1.23-1.25]. Within 44% of strongest relative associations (Rank 618 of 1419 outcomes with >100 events). | Fourth-largest crude RD of any outcome of 2.05/1,000 person-years. | Slight attenuation to 1.19 [1.18-1.20] (4.23% change). Hypertension in primary care is available in adjustment set. | HRs similar across cohorts: 1.21 [1.12-1.31] (<18), 1.24 [1.23-1.25] (any-age), 1.25 [1.23-1.25] (18 +), 1.24 [1.23-1.25] (40 +). | Attenuated in ever hospitalised cohort to 1.17 [1.16-1.18] (adjusted to 1.14 [1.13-1.15]). Larger attenuation of crude HR to 1.08 [1.07-1.09] (adjusted to 1.06 [1.05-1.07]) when excluding non-consulters. | Associations with I11 Hypertensive heart disease 1.21 [1.10-1.33], I12 Hypertensive renal disease 1.29 [1.26-1.33], and I13 Hypertensive heart and renal disease 1.30 [1.13-1.49]. Previous primary-care results | Phecode 401.1 "Essential hypertension" covers I10-I15.9 but has the same crude HR as I10 of 1.24 [1.23-1.25]. | Very common outcome with moderately strong relative association that is largely attenuated in analyses accounting for consultation behaviour or differences in baseline comorbidities. |

**Table 1 (continued) | Example associations**

| Disease | Relative strength of association (Crude hazard ratios from any-age cohort; see Figs. 3, 4 and Supplementary Table 2, 3) | Absolute strength of association (Crude rate differences from any-age cohort; see Figs. 2, 4 and Supplementary Table 2) | Adjusted results (Primary care recorded comorbidity adjusted results; see Supplementary Table 1 for recorded comorbidities; see Supplementary Table 2, 3) | Results by age (Results from <18, 18+, and 40+ cohorts; <18 cohort: exposed individuals met the eczema definition before their 18th birthday; 18+ / 40+ cohorts: individuals at least 18/40 years; see Supplementary Tables 2, 3) | Results from sensitivity analyses (Results from ever hospitalised cohort and excluding non-consulters; see Supplementary Tables 3, 4) | Related outcomes and primary-care results (if available) see Supplementary Tables 2, 4 for outcomes, and Supplementary Table 10 for results from previous primary-care study) | Phecodes (see Supplementary Table 6 for phecode-mapped outcomes) | Summary |
|---|---|---|---|---|---|---|---|---|
| | | | | | | for hypertension were similar (HR 1.11). | | |
| C81 Hodgkin lymphoma and C84 Mature T/ NK cell lymphomas | Crude HRs of 1.71 [1.54-1.90] for C81 Hodgkin lymphoma, 2.44 [2.11-2.83] for C84 Mature T/NK cell lymphomas. Within 4% and 2% of strongest relative associations, respectively (Ranks 55 and 20 of 1419 outcomes with >100 events). | All with small RD between 0.00 and 0.04/1,000 person-years. | Slight attenuation to 1.64 [1.48-1.83] (3.87% change) for C81. Minimal attenuation to 2.41 [2.08-2.80] (1.16% change) for C84. Adjustment set includes Hodgkin and non-Hodgkin lymphomas. | HR for C81 similar in any-age and 18+ cohort but attenuated to 1.62 [1.42-1.85] in 40+ cohort, and 1.41 [1.11-1.79] in <18 cohort. HR for C84 larger in other cohorts (e.g., 2.71 [2.33-3.15] in 18+ cohort), but <100 events in <18 cohort. | Attenuated in ever hospitalised cohort to 1.46 [1.30-1.65] (adjusted to 1.43 [1.26-1.61]). Weaker attenuation of crude HR to 1.51 [1.30-1.77] [adjusted to 1.47 [1.26-1.72] when excluding non-consulters. | HRs for C82 Follicular lymphoma (1.16 [1.04-1.29]), C83 Non follicular lymphoma (1.24 [1.17-1.32]), and C85 Other [...] Hodgkin lymphoma (1.33 [1.25-1.40]) are much smaller than HRs for C81 and C84. Previous primary-care results for Hodgkin's lymphoma were similar (HR 1.85). | Phecode 201 "Hodgkin's disease" covers C81. C84 is combined with other lymphomas in phecode 202.2 "Non-Hodgkins lymphoma" (HR 1.32 [1.27-1.38]). 229.1 "Benign neoplasm of lymph nodes" has an HR of 1.66 [1.22-2.26]. | Very rare outcomes, but some of the largest strengths of associations observed. Strength of association relatively consistent across age cohorts and remained strong in sensitivity analyses. In <18 cohort some attenuation seen for C81 and C84 too rare to observe. |
| E10 Type 1 diabetes mellitus | Crude HR of 1.23 [1.19-1.26]. Within 47% of strongest relative associations (Rank 665 of 1419 outcomes with >100 events). | RD of 0.09/1,000 person-years. | Attenuation to 1.15 [1.12-1.19] (6.15% change). Adjustment set includes diabetes mellitus (including all types). | HR somewhat increased in 18+ and 40+ cohorts, but no association in <18 cohort (0.97 [0.91-1.04]). | Attenuated in ever hospitalised cohort to 1.14 [1.11-1.18]. Adjusted HR at the null when excluding non-consulters (1.01 [0.97-1.05]). | Previous primary-care results were similar (adjusted HR of 1.87 for diabetes mellitus in any-age cohort) but included other types of diabetes. | Phecode 250.1 "type 1 diabetes" covers E10 Type 1 diabetes mellitus. | Adjusted HR at the null when excluding non-consulters, and crude HR at the null in the <18 cohort (which would cover the usual age of onset of type 1 diabetes) suggest bias in estimates from adult cohorts. |

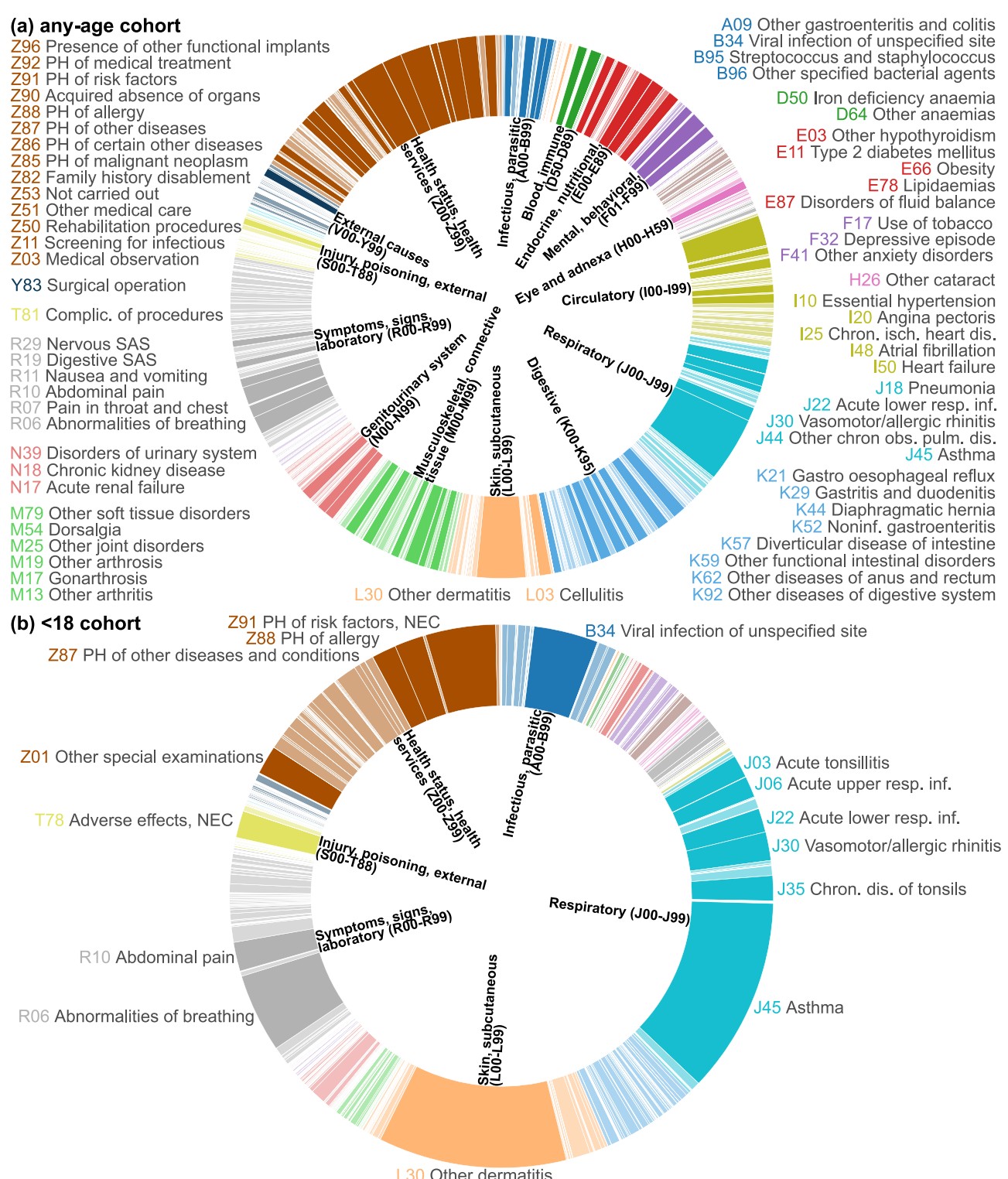

**Fig. 2 | Excess rate for all outcomes.** Excess rate (segment width) as the proportion of the sum of all excess rates from crude results from (a) the any-age cohort (i.e., exposed individuals met the eczema definition at any age), (b) the <18 cohort (exposed individuals met the eczema definition before their 18th birthday). Each outcome is defined as a 3-character ICD-10 code and its descendants. Labels and increased opacity of segments are shown for outcomes with the largest individual excess rate estimates that cumulatively make up 50% of the total excess rate. Outcomes with ≤100 events or an excess rate in the control group rather than the eczema group are not shown. Outcomes are arranged clockwise in order of ICD-10 code. Outcome labels are abbreviated. PH = Personal history, SAS = signs and symptoms, NEC = not elsewhere classified. Source data are provided as a Source Data file.

Based on the strength of association, consistency across cohorts and analyses, alternative explanations for observed effects, novelty, and potential implications for clinical practice, some of the most important findings from this study are: 1. Strong evidence for associations with ophthalmologic outcomes, for which there is currently insufficient awareness, including rarer and more serious keratoconus and cataracts; 2. The strongest signal for an outcome not typically regarded as a direct complication of atopic eczema or atopy

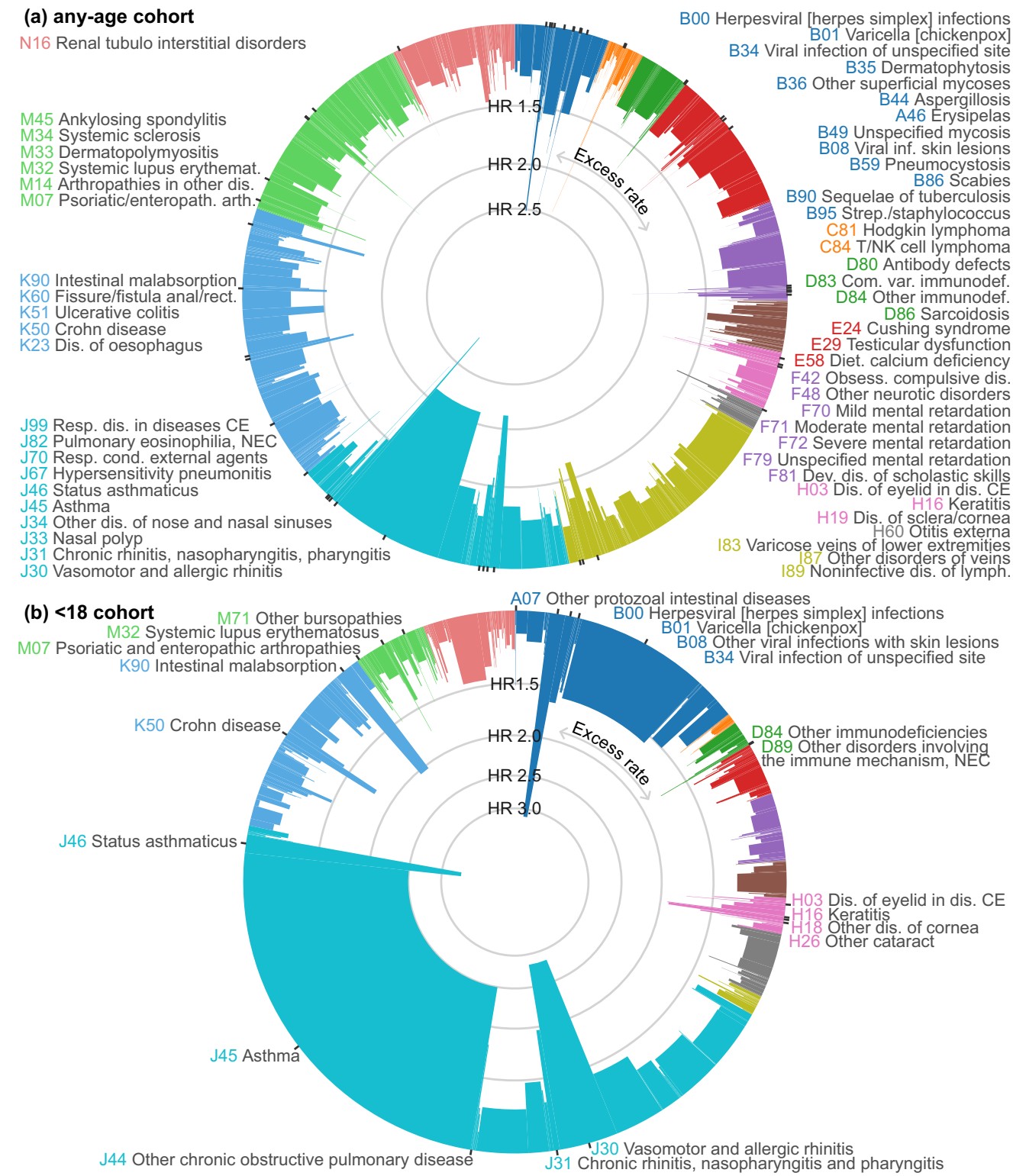

**Fig. 3 | Hazard ratios and excess rate for all outcomes (excluding chapters L, Q-Z).** Hazard ratios (segment height) and excess rate (segment width) as the proportion of the sum of all excess rates from crude results from (**a**) the any-age cohort (i.e., exposed individuals met the eczema definition at any age), (**b**) the < 18 cohort (exposed individuals met the eczema definition before their 18th birthday). Each outcome is defined as a 3-character ICD-10 code and its descendants.

Excludes chapters L & Q-Z. Labels and ticks are shown if the hazard ratio > 1.5. Outcomes with ≤ 100 events or an excess rate in the control group rather than the eczema group are not shown. Outcomes are arranged clockwise in order of ICD-10 code. Outcome labels are abbreviated. CE = classified elsewhere, NEC = not elsewhere classified. Source data are provided as a Source Data file.

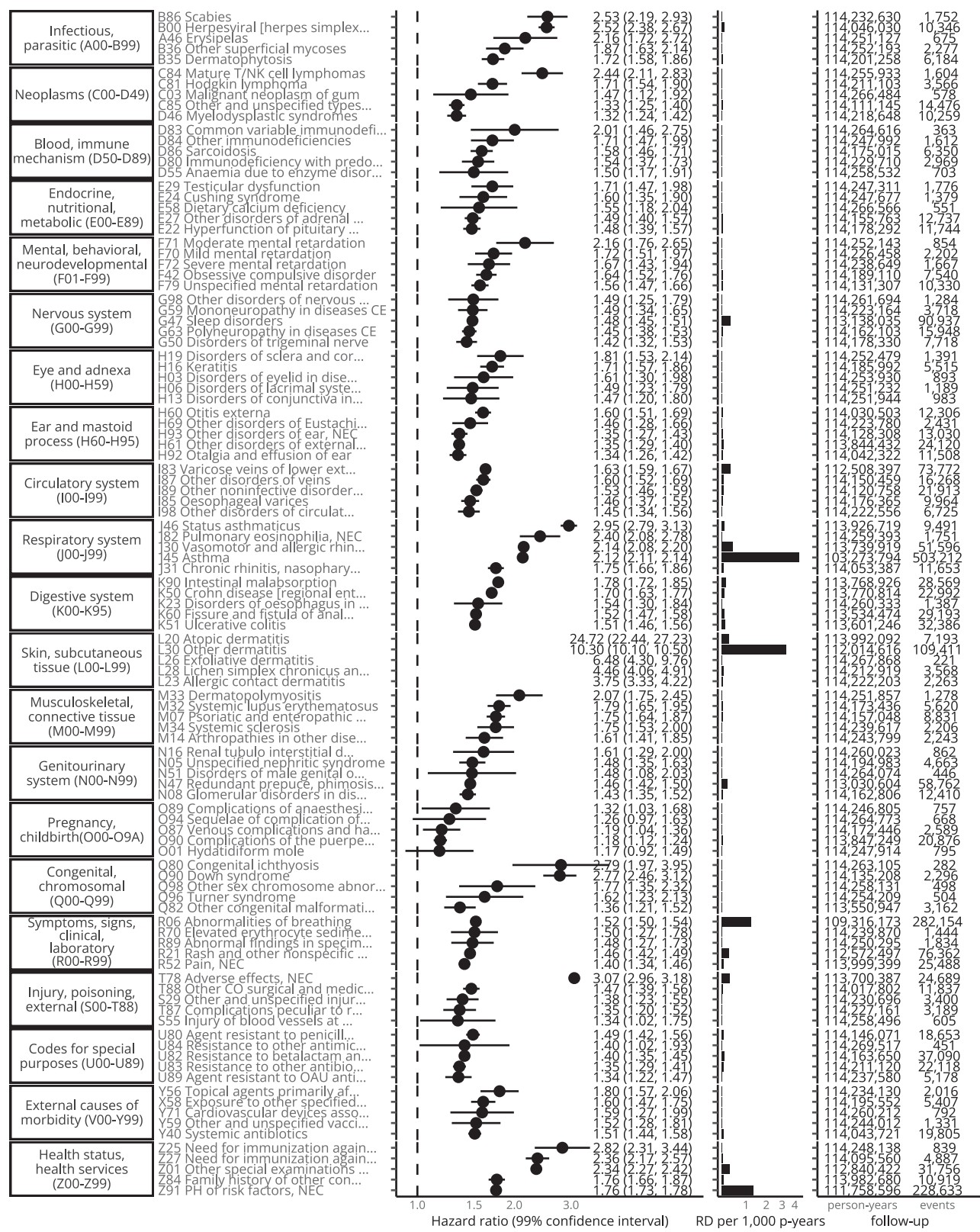

**Fig. 4 | Five strongest associations per ICD-10 chapter (any-age cohort).** Five strongest associations between eczema and outcomes for each ICD-10 chapter (with > 100 events) form the any-age cohort. Hazard ratios (points) with 99% confidence intervals (lines) from Cox regression and crude rate difference per 1000 person-years (RD per 1000 p-years) (bars). Source data are provided as a Source Data file.

was for Crohn's disease; 3. Reassuringly, for people who had eczema in childhood, any risks of outcomes were comparatively small compared to outcomes already informing atopic eczema diagnosis (e.g., asthma, allergy, skin infections, conjunctivitis).

The use of routinely collected electronic health records (EHR) data comes with several strengths. Large sample size enabled follow-up of even rare outcomes. CPRD Aurum has been found to be representative of the general population of England in terms of age, sex,

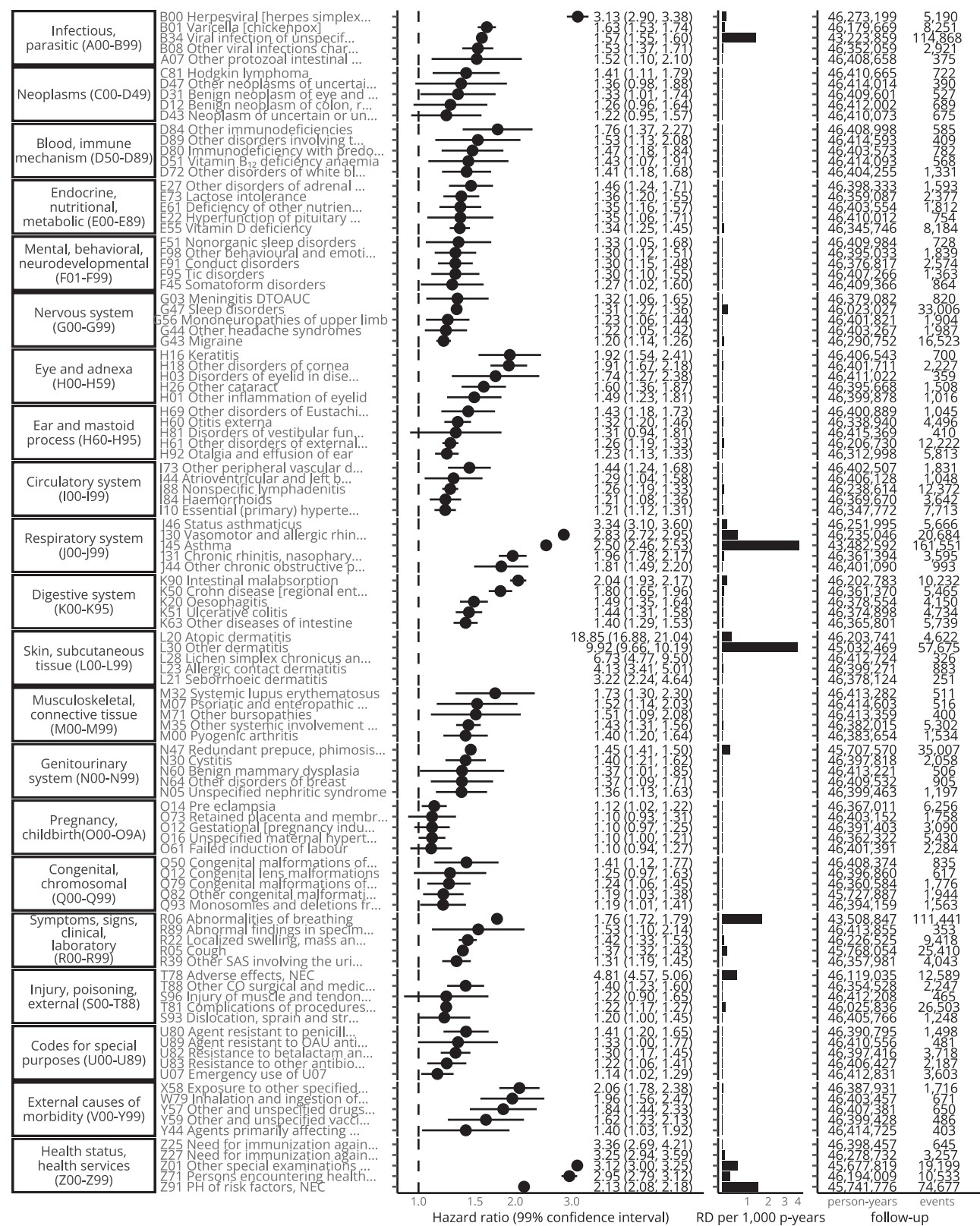

**Fig. 5 | Five strongest associations per ICD-10 chapter ( < 18 cohort).** Five strongest associations between eczema and outcomes for each ICD-10 chapter (with > 100 events) from the < 18 cohort. Hazard ratios (points) with 99% confidence intervals (lines) from Cox regression and crude rate difference per 1000 person-years (RD per 1000 p-years) (bars). Source data are provided as a Source Data file.

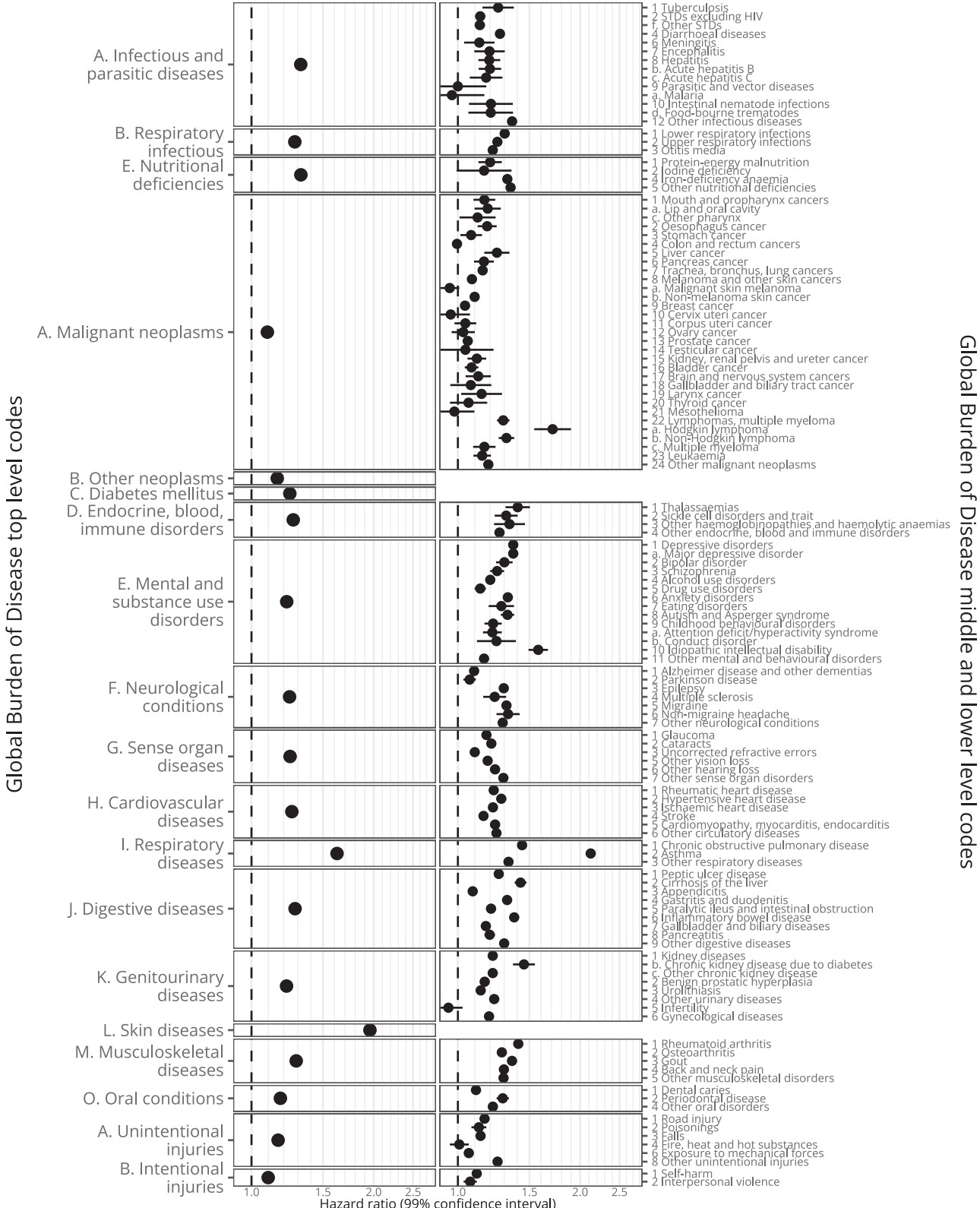

**Fig. 6 | Global Burden of Disease code-mapped results (any-age cohort).** Hazard ratios (points) with 99% confidence intervals (lines) from Cox regression. Excludes outcomes with ≤ 200 events and Congenital anomalies, Maternal conditions, and Neonatal conditions. The number of events is provided in Supplementary Table 5. Source data are provided as a Source Data file.

geographical spread and deprivation (albeit it may not necessarily be generalisable to other settings)[6]. Using primary care data to define the exposure of eczema, we were likely able to avoid some of the selection biases that can occur when studying hospital-based populations[7].

The use of ICD-10 coded hospital records allowed us to systematically and comprehensively study all hospital-recorded diseases, a previously underutilised strength of electronic health records. Ascertaining outcomes in hospital records may also make

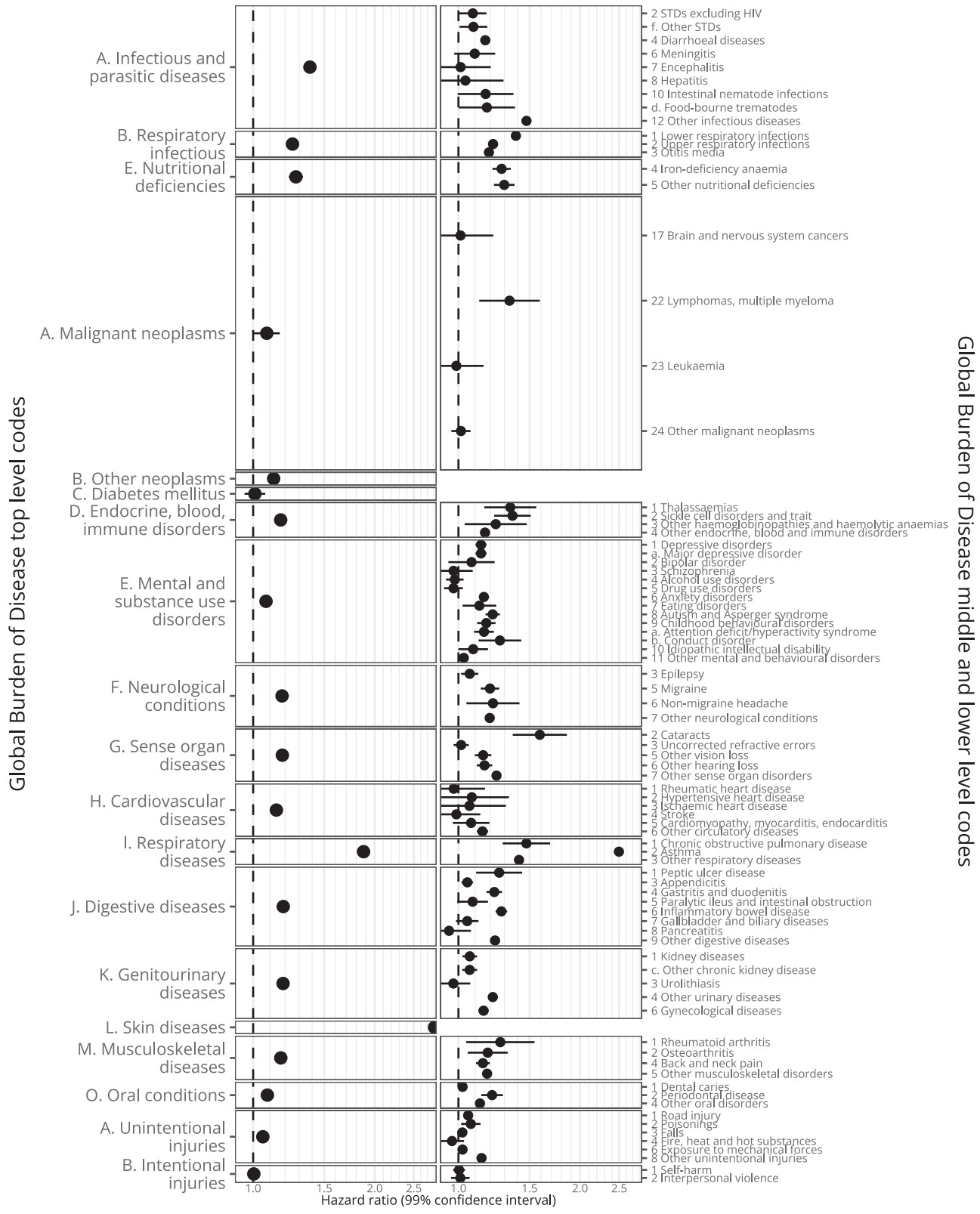

**Fig. 7 | Global Burden of Disease code-mapped results (< 18 cohort).** Hazard ratios (points) with 99% confidence intervals (lines) from Cox regression. Excludes outcomes with ≤ 200 events in the exposed and Congenital anomalies, Maternal conditions, and Neonatal conditions. The number of events is provided in Supplementary Table 5. Source data are provided as a Source Data file.

observed differences less likely to be due to ascertainment bias (e.g., people with eczema may consult their GP more frequently because of their eczema, but are unlikely to be admitted to hospital more frequently). The use of different ICD-10 mappings is another

strength of this study. Firstly, the use of GBD codes gives a more compact overview across diseases, and results will be comparable to those from the Global Burden of Disease Studies. Secondly, the use of phecodes complements the use of category-level ICD-10

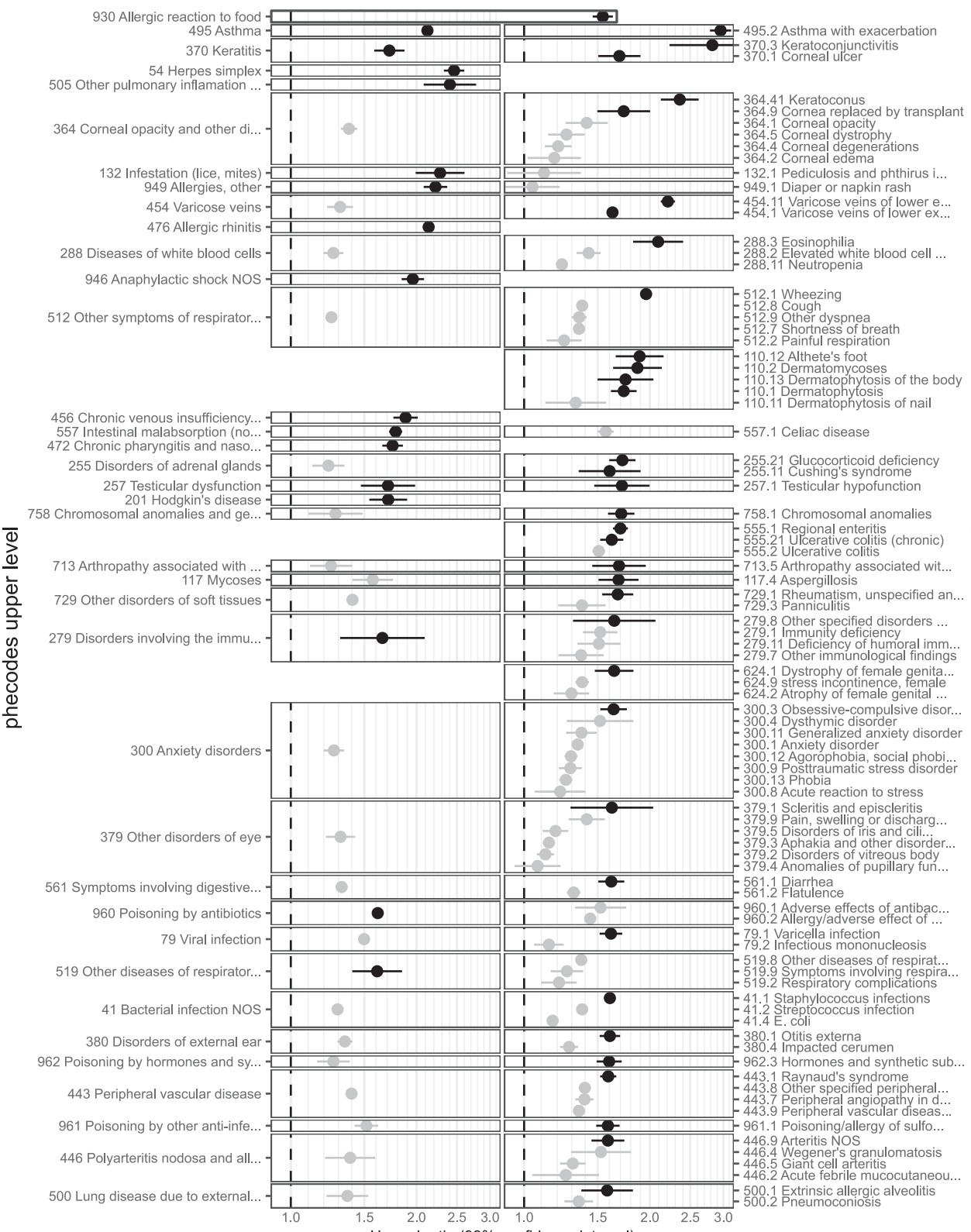

**Fig. 8 | 40 strongest associations from phecode-mapped results (any-age cohort).** Hazard ratios (points) with 99% confidence intervals (lines) from Cox regression. The 40 phecodes with the largest hazard ratios are shown (in black), together with all codes that have the same parent code (in grey). Excludes outcomes with ≤ 200 events in the exposed. The number of events is provided in Supplementary Table 6. Source data are provided as a Source Data file.

codes. While many diseases of interest correspond well to category-level ICD-10 codes, for some, phecodes offer a more clinically useful definition. However, the use of unmapped category-level ICD-10 codes has the advantage of giving a complete

picture of all hospital records, as well as outcomes being non-overlapping.

Previous hypothesis-free studies exploring eczema outcomes employed study designs not accounting for the temporality of

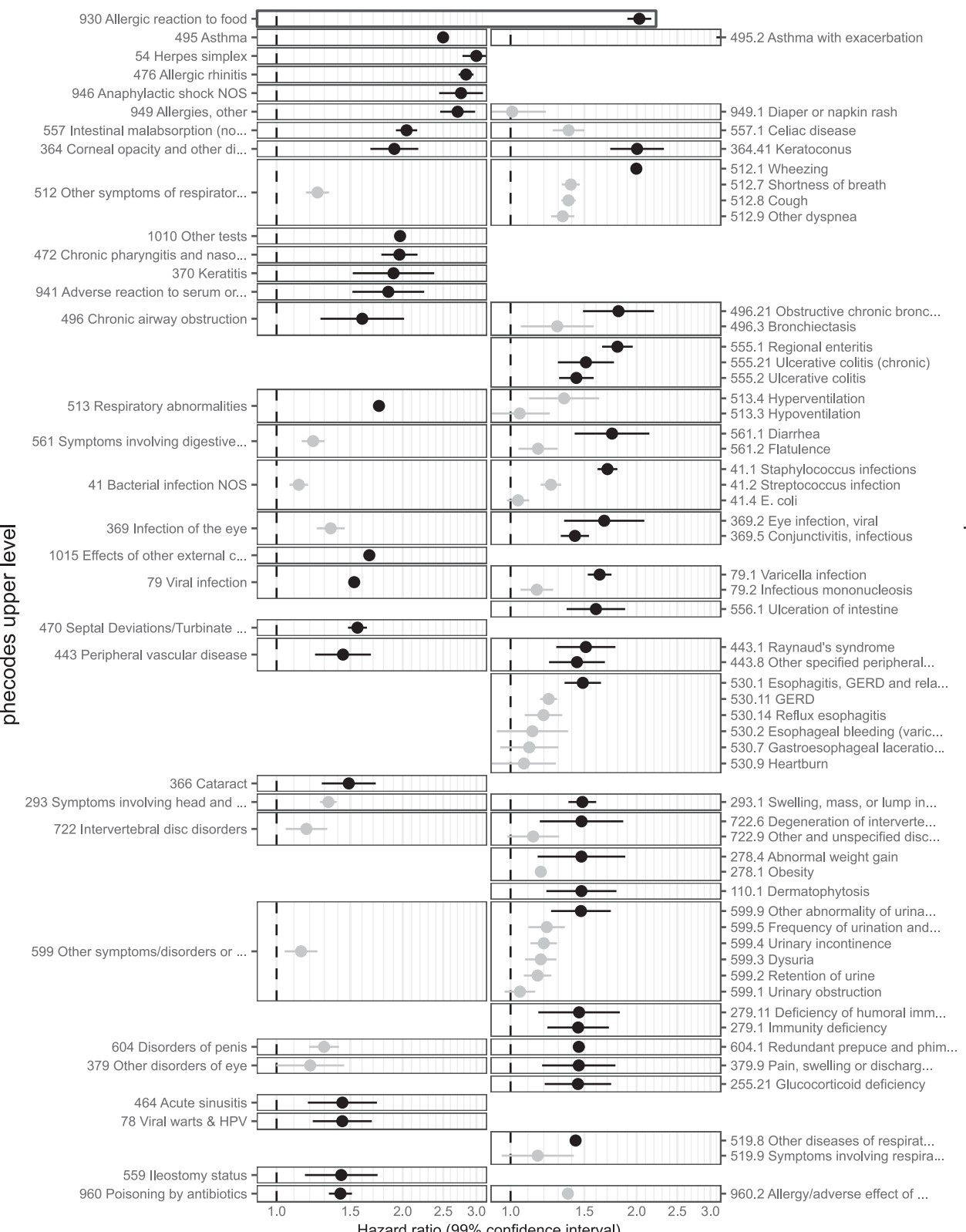

**Fig. 9 | 40 strongest associations from phecode-mapped results (<18 cohort).** Hazard ratios (points) with 99% confidence intervals (lines) from Cox regression. The 40 phecodes with the largest hazard ratios are shown (in black), together with all codes that have the same parent code (in grey). Excludes outcomes with ≤ 200 events in the exposed. The number of events is provided in Supplementary Table 6. Source data are provided as a Source Data file.

associations and low granularity of outcomes (e.g., on ICD-10-chapter level)[8]. Our study offers high granularity of outcomes using a study design that could be employed in hypothesis-testing studies. The evidence created through this approach is a more useful starting point

for future studies, allowing these to focus on outcome-specific considerations.

This study has limitations. Results may be biased by individuals already having the outcome of interest before it is captured in hospital

records. We excluded individuals who already had a record for the outcome under analysis before the index date and performed sensitivity analyses requiring previous hospital admission. These measures are, however, unlikely to account for all instances of individuals already having the relevant outcome before they develop eczema. For some outcomes, adjustment for comorbidities recorded in primary care at baseline can partially account for outcomes pre-dating eczema (e.g., a large attenuation is seen for asthma in comorbidity-adjusted [including for asthma] results).

A limitation of this large-scale study is a lack of outcome-specific confounding adjustment strategies. While our covariate set of primary care assessed comorbidities was chosen based on the existing literature to include the most relevant comorbidities of eczema, there may still be residual confounding for any outcomes.

Information bias, including misclassification of exposure and outcome as well as their timings, needs to be considered, and can vary for each individual outcome. For example, outcomes that can present with a rash might be initially misdiagnosed as eczema until they get the correct diagnosis; a likely explanation for many other skin disease outcomes, including psoriasis, but also other outcomes presenting with rash, such as systemic lupus erythematosus and certain lymphomas[9]. Outcome status and timing may also be misclassified, both through factors relating to disease progression (e.g., a long lead time before diagnosis), as well as factors relating to the use of hospital admission data to capture outcomes (e.g., outcomes not requiring hospital admission only being recorded when individuals are admitted to the hospital for a different reason).

An important finding of our study was the stark difference between the <18 cohort (i.e., exposed individuals met the eczema definition before their 18th birthday) and the other cohorts. In the <18 cohort, eczema was associated with fewer outcomes, absolute risk increases were less distributed across organ systems, and several outcomes found to be associated in the other cohorts were not associated in the <18 cohort (Supplementary Table 8). Several factors may explain these differences. Firstly, follow-up extending a maximum of 25 years may be too short to observe outcomes occurring most commonly in older age, or outcomes for which the risk may only increase after many years of living with eczema (potentially due to chronic systemic inflammation)[10].

Secondly, the <18 cohort consists mainly of incident eczema rather than incident and prevalent eczema (as do the other cohorts), which may better account for the temporality of associations. For example, despite excluding individuals with hospital records for the outcome before the index date, there is a strong association with Down's syndrome (a congenital condition that is guaranteed to have occurred before eczema and is likely to increase the risk of eczema)[11] in all cohorts except the <18 cohort, where there was no association.

Thirdly, there may be increased confidence in the diagnosis of atopic eczema in childhood, as opposed to other forms of dermatitis (irritant, contact, varicose eczema, etc.). An example of this may be the very strong association with I83 "Varicose veins [...]" in the any-age cohort, as opposed to no association in the <18 cohort: the any-age cohort may include individuals that have varicose eczema rather than atopic eczema. However, other outcomes that are likely to be strongly associated due to misclassified eczema remain strongly associated in the <18 cohort, including psoriasis (2.04 [1.84−2.26]) and systemic lupus erythematosus (1.73 [1.30−2.30]).

Fourth, the direct effects of eczema as opposed to the effects of eczema treatments, such as corticosteroids, are likely more difficult to distinguish in adult cohorts (e.g., illustrated by the strong association with Cushing syndrome).

Eczema most commonly occurs first in childhood and often improves in adulthood, but can also persist or occur at older ages[12]. All cohorts include a mix of individuals whose eczema has gotten better or

even disappeared over time, as well as individuals who continue to have eczema throughout their life.

Chapters including respiratory, eye, and digestive system conditions stand out in that the outcomes with the largest HRs were larger in the <18 cohort than the any-age cohort. While unsurprising for atopic and allergic outcomes (asthma, allergic conjunctivitis/rhinitis, allergies), it is also the case for other outcomes including cataract, intestinal malabsorption (which includes coeliac disease), and inflammatory bowel conditions (Crohn's disease and ulcerative colitis). We highlighted inflammatory bowel conditions as outcomes of particular interest in our previous primary-care study[3].

Generally, results from the 18+ and 40+ cohorts were similar to those from the any-age cohort, however, where differences occur, they may suggest differences in outcome risk by age. For example, atopic and allergic outcomes (food allergy, asthma, allergic rhinitis) and viral infections (herpes simplex, varicella) had much smaller hazard ratios in the 40+ cohort than the any-age cohort, which seems plausible given the usual ages of onset for these conditions (Supplementary Table 9).

Several diseases of the eye had strong associations with eczema. While ophthalmologic complications have long been recognised in dermatology (e.g., as diagnostic features of atopic dermatitis)[13], there have been few epidemiological studies and limited awareness in clinical practice[14]. Our findings highlight eye outcomes, including the more serious keratoconus and cataracts, as likely to be key complications of eczema. New drugs to treat eczema (e.g., dupilumab) have been found to increase the risk of ocular adverse reactions (e.g., dry eye, conjunctivitis)[15], making it important to understand the underlying risk of eye conditions in people with eczema.

The association with Crohn's disease is notable in several ways. In all cohorts and analyses, it has the largest relative and absolute risk increase of all outcomes that would not be considered direct complications of eczema and atopy and not likely to be associated due to eczema misdiagnosis. The association is stronger than for ulcerative colitis, which itself was a strong association, matching findings from previous studies[16]. Our findings highlight, for the first time, Crohn's disease as one of the most strongly associated in the context of all hospital-recorded outcomes. While Crohn's disease can have cutaneous manifestations, these are not likely to be misdiagnosed as eczema, and a Mendelian randomisation study (which are less prone to reverse causation) found a unidirectional effect from eczema to inflammatory bowel disease[17]. Another possibility may be an association with atopy more generally, e.g., as has been found with asthma[18].

The only association with a digestive system outcome stronger than Crohn's disease was for intestinal malabsorption. Phecode-mapped results suggest that the risk for non-coeliac food intolerances is larger than for coeliac disease. Again, reverse causation may partially explain this association, with dermatitis herpetiformis (a blistering, itchy rash) being a manifestation of coeliac disease[19]. The next strongest digestive system association were oesophagitis (possibly related to atopy-related eosinophilic oesophagitis, for which awareness was previously low)[20], irritable bowel syndrome, and gastritis.

Several other strongly associated outcomes are likely to be associated due to reverse causation, with cutaneous manifestations that may appear like eczema preceding diagnosis. These include various other skin conditions, immunodeficiencies[21], and lymphomas[9]. For Hodgkin's lymphoma, our findings (including smaller HR in the <18 cohort with clearer atopic eczema diagnosis), together with findings from previous studies[22] cannot fully prove or discount reverse causation. The outcome was very rare and therefore unlikely to be of importance for the management of eczema (but possibly relevant in explaining the aetiology of Hodgkin's disease)[23].

Further outcomes may be associated as downstream complications of diseases that could be misdiagnosed as eczema, for example, arthritis may be due to psoriasis. For other outcomes with large

relative and absolute increases in risk, e.g., for phimosis, explanations may yet still be unclear.

Other outcomes are likely to be direct manifestations or complications of eczema, including genital or nipple eczema, infections of the ear and other locations. Sleep disorders, likely related to nighttime itch, is one of the more common moderately strongly associated outcomes, with previous research and clinical practice agreeing[24].

For anxiety and depression, our findings match those from previous studies in UK primary care data, suggesting only small relative increases in risk, which, however, could still be important given anxiety and depression are common[3,25]. Obesity had a similar relative and absolute risk increase. For several other outcomes, our findings would only suggest small increases in risk, which may be explained by residual confounding or bias. For some outcomes, small increases in risk may be explained by risk only being elevated in people with more severe eczema, e.g., as was suggested in our previous study for cardiovascular conditions and fractures[3]. For several outcomes, e.g., most cancers, we did not find any increase in risk.

We previously assessed associations with 71 outcomes recorded in primary care using the same cohort definitions (i.e., any-age, <18, 18+, 40+) (without requiring eligibility for HES linkage). In Supplementary Table 10 we compare our results against these previous results (for the 69 outcomes we were able to identify a corresponding ICD-10 code or phecode), using the same adjustment set. Results were generally similar (previous vs this study e.g., food allergy HR [99%CI] 4.03 [3.95–4.11] vs 930 Allergic reaction to food 4.37 [4.11–4.63]; Diverticular disease 1.17 [1.16–1.18] vs K57 Diverticular disease of intestine 1.17 [1.16–1.19]; Prostate cancer 1.01 [0.99–1.03] vs C61 Malignant neoplasm of prostate 1.04 [1.01–1.06]). Results were also generally similar for outcomes unlikely to be primary reasons for hospital admission (previous vs this study, e.g., Hypertension 1.11 [1.10–1.12] vs 1.19 [1.18–1.20]), suggesting these diagnoses are recorded during hospital admissions, even if they are not the primary reason for admission.

Our findings are generally reassuring and seem plausible, judging from clinical practice: in a cohort of people who first had eczema in childhood, besides the already well-known outcomes, including atopic and allergic conditions and infections, there is little evidence for considerable increases in risks for most other diseases. Some other strongly associated outcomes have potential alternative explanations relating to eczema misdiagnosis (e.g., lymphomas). As in our previous study in primary care, inflammatory bowel diseases, particularly Crohn's disease, emerge as outcomes of interest, for which the evidence from other studies using large routinely collected data[16], robustly verified cohorts from specialist atopic dermatitis clinics[26], birth cohorts[27], and Mendelian randomisation studies[17] all support this association. Future research should take into account evidence from these various sources, identify open questions that still cast doubt on this association, aim to explain mechanisms (such as treatments, genetics or shared underlying immune dysfunction), to ultimately help decide whether screening individuals with eczema for inflammatory bowel disease could improve outcomes.

For diseases of the eye, including the more rare and serious cataract and keratoconus, while the evidence from other epidemiological studies is sparse, the implications may be clearer. Recent guidelines for dermatologists in the Nordic region, including guidance on when to consider referrals to ophthalmology, may be applicable more generally[14].

We may consider results from the <18 cohort as most accurate, given the clearer diagnosis of atopic eczema in childhood, as well as the avoidance of time-related biases (through using a cohort with primarily incident eczema). However, given a maximum follow-up time of 25 years, it is also useful to examine cohorts of adults, including both incident and prevalent eczema to assess risks across the life course. Despite a larger potential for bias, results from these adult cohorts

suggest increased rates of hospital-recorded diagnoses across multiple organ systems. To know if these outcomes could be prevented, e.g., through better treatment of eczema, we need further research considering factors such as eczema severity, treatments, and outcome-specific considerations.

In conclusion, our study, to our knowledge, is the first to offer both a big-picture view of risks faced by people with eczema across the entire disease spectrum, as well as allowing detailed inspection of each individual association. Besides filling in gaps in knowledge, the comparable estimates facilitate prioritisation in clinical practice, including through informing guidelines (e.g., on comorbidity symptom screening or multidisciplinary referrals). Incorporating all diseases and allowing for comparisons make our findings a hypothesis-generating tool for future studies on any eczema-associated disease outcome.

## Methods

### Study population
Deidentified linked data were provided by the CPRD[28]. We used an algorithm to identify individuals with eczema (at least one record of an eczema diagnosis code and at least two prescription records for eczema therapies [emollients, topical glucocorticoids, topical calcineurin inhibitors, oral glucocorticoids or systemic immunosuppressants] on two separate days). An analogous algorithm has been previously validated in UK primary care data and was found to have a positive predictive value of 86%[29]. We included individuals in the eczema cohort from the latest of: (1) Date they met the eczema definition; (2) One year after practice registration (to allow us to reliably capture baseline health status); (3) Study start (April 1, 1997); and (4) 18th (18+ cohort) or 40th birthday (40+ cohort), or no age limitation (any-age cohort). For the <18 cohort, exposed individuals had to meet the eczema definition before their 18th birthday. Meeting the eczema definition could occur before individuals became eligible (i.e., individuals with both new and existing eczema were included, a recommended approach for relapsing conditions like eczema to better assess longer-term effects of an exposure)[30].

Eczema exposed individuals were matched (without replacement) to up to 5 unexposed individuals with at least 1-year prior registration, on age (2-year calliper), sex, and general practice in calendar date order. The index date for comparators was set to the index date of the exposed individual they were matched to. Comparators were censored on the day they met the eczema definition themselves and could then be re-matched as exposed individuals. Individuals were followed until the date of specific outcome (depending on the outcome under investigation), or until they were censored (death, left practice, or for comparators, when they met the eczema definition). For each outcome-specific analysis, individuals who had the outcome before their index date were excluded (Fig. 1).

### Outcomes
We assessed outcomes recorded in any diagnostic position (i.e., primary or any other reason for admission or medical history) in Hospital Episode Statistics (HES) Admitted Patient Care (APC) data. To systematically learn about risks across the full health spectrum we used multiple disease classification systems, each with different strengths and limitations: (1) 3-character ICD-10 codes (including all descendant codes) (2,058 outcomes); (2) phecodes (a high-throughput phenotyping used to define clinically meaningful diseases and conditions)[31] (1593 outcomes)[4]; and (3) Global Burden of Disease codes at different levels of granularity from a technical paper on the methods for the Global Burden of Disease studies (201 outcomes)[5].

### Statistical analysis
We used Cox proportional hazards regression, stratifying on matched set, to estimate hazard ratios (HRs) and 99% confidence intervals (99%CI) for the association of eczema with each outcome. For

each analysis, we estimated crude HRs (implicitly adjusted through matching on age, sex and general practice, and calendar time, as comparators entered the cohort on the same day as exposed individuals). To account for baseline differences in comorbidities and to allow a comparison with adjusted-results from our previous primary care study, we also estimated comorbidity-adjusted HRs (additionally adjusted for history of 71 conditions assessed in primary care from the following disease categories: Atopic and allergic, Immune mediated, Mental health and substance use, attention deficit hyperactivity disorder and autism, Cardiovascular, Metabolic, Bone health, Skin infection, Cancer, Neurological, Digestive system, Liver), as was done in our previous study[3].

Since comparators were matched, we estimated rates of outcomes expected in the absence of eczema as the rate in people with eczema multiplied by the inverse of the corresponding hazard ratio.

To account for multiple testing, we reported wider 99% instead of the usual 95% confidence intervals and defined a Bonferroni corrected significance threshold (with 2058 ICD-10 categories considered, estimates are considered significant under Bonferroni correction with a p-value less than $0.01/2058 = 0.000005$).

### Sensitivity analyses

We conducted two types of sensitivity analyses. Firstly, to account for individuals already having the outcome of interest at the index date, but no hospital record for it (yet), we included only individuals that had been hospitalised at least once in the year before the index date.

Secondly, to address consultation bias or differential health seeking behaviour we excluded "non-consulters", i.e., individuals who did not have any of four common primary care records (9N11.00 - Seen in GP's surgery, 22 A..00 - Body weight, 4....00 - Laboratory test, 246..00 - O/E - blood pressure reading) in the year prior to index date (as was done in our previous study)[3].

### Pipeline

For all 2058 outcomes based on 3-character ICD-10 codes, we ran analyses for all four cohorts (any-age, <18, 18+, 40+) and an additional cohort for sensitivity analyses (ever-hospitalised), two models (crude and comorbidity adjusted), and with and without exclusion of non-consulters (sensitivity analysis), resulting in a total of 41,160 combinations. For all 1593 phecodes and 201 Global Burden of Disease (GBD) codes, we ran crude and adjusted analyses for all four cohorts and the ever-hospitalised cohort (15,930 for phecodes, 2010 combinations for GBD codes). We used R version 4.3.1 and organised the research pipeline using the targets R package, ensuring reproducibility of the computationally expensive pipeline[32].

### Role of the funding source

The study funder had no role in study design, data collection, data analysis, data interpretation, or report writing.

### Ethics

The study was approved by the London School of Hygiene & Tropical Medicine Research Ethics Committee (Reference number: 29781). This study is based on data from the CPRD obtained under license from the U.K. Medicines and Healthcare products Regulatory Agency. The data are provided by patients and collected by the National Health Service (NHS) as part of their care and support. The study was approved by the CPRD Independent Scientific Advisory Committee (Protocol reference number: 23_002665). Individual patient consent is not required or possible since CPRD provides deidentified data.

### Reporting summary

Further information on research design is available in the Nature Portfolio Reporting Summary linked to this article.

## Data availability

Data supporting the findings of this study are available in the article and its Supplementary Information. The data underlying this article is provided by the UK CPRD electronic health record database, which is only accessible to researchers with protocols approved by the CPRD's independent scientific advisory committee. Data access may incur a cost, and further details can be found here: https://www.cprd.com/data-access. Source data for graphs and Supplementary Tables are provided with this paper. All Interactive Figures and Tables can be found in the dashboard available at https://github.com/julianmatthewman/Eczema_hospital_outcomes_public. Source data are provided in this paper.

## Code availability

All analysis code and codelists used for this study are available at https://doi.org/10.5281/zenodo.17433428.

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

## Acknowledgements

This work uses data provided by patients and collected by the NHS as part of their care and support. This work was funded by a Wellcome Trust Senior Research Fellowship in Clinical Science (205039/Z/16/Z) awarded to Sinéad M. Langan. This work was supported by Health Data Research UK, an initiative funded by UK Research and Innovation, Department of Health and Social Care (England) and the devolved administrations, and leading medical research charities. Krishnan Bhaskaran is funded by a Wellcome Senior Research Fellowship (220283/Z/20/Z). The views expressed in this publication are those of the author(s) and not necessarily those of the NIHR, NHS or the UK Department of Health and Social Care.

## Author contributions

J.M. contributed to Conceptualisation, Data curation, Formal Analysis, Funding acquisition, Investigation, Methodology, Project administration, Resources, Software, Supervision, Validation, Visualisation, Writing – original draft, and Writing – review & editing. S.M.L. contributed to Methodology, Supervision, Validation, and Writing – review & editing. J.M., A.S., K.B., A.R., S.D., K.E.M., and S.M.L. contributed to Methodology and Writing – review & editing.

## Competing interests

Julian Matthewman, Anna Schultze, Spiros Denaxas, Krishnan Bhaskaran, and Amanda Roberts have no competing interests to report relating to the findings. Sinéad M Langan is a co-investigator in a consortium with industry and multiple academic partners (BIOMAP-IMI.eu) but is not in receipt of industry funding. Kathryn E Mansfield reports individual consulting fees from AMGEN.

## Patient and public involvement

This research was informed by AR, an independent patient partner and co-author of this work.
