## [Transparent Peer Review file · Nature Communications]

Mapping risks of health conditions in people with atopic eczema in English primary care and hospital data

Corresponding Author: Dr Julian Matthewman

Version 0:

Reviewer comments:

Reviewer #1

(Remarks to the Author)

A well-written and very informative paper. The authors used public records in UK to test associations of atopic dermatitis with diseases across the full health spectrum. They clearly state the limitations and potential use of their findings.

(Remarks on code availability)

I have not installed and run the code.

Reviewer #2

(Remarks to the Author)

This is a comprehensive study, one of the largest and most credible yet conducted on AD comorbidities.

The authors have a long history of credibility in these investigations and have contributed greatly to this field.

The study builds on prior work, with important new findings on eye disease, confirmations and clarifications of the effect size of prior associations such as IBD, first identified in 2006 by Weidinger (PMID: 26253344).

The data compiled here will have great utility in driving further work to understand the extensive co morbidities of Atopic Dermatitis.

The methodologies employed are sound: the databases are very reliable, statistical methodologies applied are appropriate, well conducted and reported

I think this work is at the threshold for Nat Comms acceptance. If I were to be critical, the work may not have many standout truly novel headline findings.

(Remarks on code availability)

I checked the github url for data completeness but did not re-run the source code. It does have a README file.

Reviewer #3

(Remarks to the Author)

General Remarks

1. Matthewman et al. made an enormous effort to provide an analysis of cohort of people with eczema compared to matched comparators in millions of people with up to 25 years of individual follow-up. It is a very elaborate article that follows the previous study from Matthewman et al. 2024 (<https://doi.org/10.1038/s41467-024-54035-1>).
2. Perhaps it could be explained in more detail what exactly the added value is in presenting the hazard ratios in the various classification systems (ICD, phecodes, GBD)?
3. In addition, it might be useful for readers to explain how the linkage between the primary care data and hospital records took place and how hospital patients were selected.
4. We were wondering whether this analysis followed a predefined protocol/analysis plan and whether the study-reporting

summary would be available (we did not discover it in the review portal).

5. Please include study design in the title so it can be detected in systematic reviews.

Abstract

6. Please include risk estimates for “large relative risk increases” for IBD and eye diseases.

Introduction

7. In our view, the introduction does not yet sufficiently explain the rationale for this enormous effort to process this large amount of data.

8. Could you please elaborate in more detail the motivation of linking data collected upon hospital admission with primary care data for the purpose of learning more about risk factors for AE (L47-49).

9. Why is ascertainment bias (L37) an issue when assessing ‘outcomes’ (as per your article this would be diagnoses or procedures in connection with secondary diseases, right?)

Results

10. The results section contains a lot of technical explanations, which in our view should be part of the methods section. This leaves little room for the key results, which readers have to somewhat ‘work out’ from the supplement tables and the dashboard.

11. Which group is referred to by ‘same cohort’ (L163/164), the ‘any age cohort’? Therefore, comparisons with previous results could be better placed in the discussion than in the results section.

12. We really liked the dashboard. It remains to be clarified whether the editors will allow this as a regular part of the article.

Discussion

13. The summary of findings is well organised, but these results are not to be found in the text of the results section, but (as mentioned above) are to be searched for by the reader in various tables.

14. It could be useful to incorporate the structure of the summary of most important findings (ophthalmologic outcomes, signals for Crohn’s disease, relatively small outcome risks comparing people with/without eczema in childhood) again for the section from L294 on.

15. Strengths and limitations have been considered in a very detailed and thoughtful way.

16. Comparison of U18-cohort with others (L250ff.): “Several factors may explain ...” it is hard to find those comparisons in the results section.

17. How realistic is it to have an even longer observation period than 25 years? The U18 cohort is already the one for which the individual observation period was probably the longest? Which secondary/concomitant diseases of AE that only occur in old age would be relevant from your point of view?

18. The age at the time of the index was significantly younger (see Table 1). Did you consider, particularly in the U18 cohort, that these individuals might have had fewer hospitalisations and therefore fewer ‘outcomes’ in general?

19. The chain of associations that the in-hospital diagnosed anaemia (L341) is a direct consequence of nutritional deficiency caused by food allergy caused/triggered by AE is somewhat speculative from our point of view.

Conclusion

20. In what way do you think your results could inform guidelines (L391)

Methods

21. Anonymised/pseudonymised data (L402-404): It might be confusing to read these two sentences, particularly for readers who are not experts of NHS data. Can patients generally opt out, so that their data will not be used for research purposes? Why should (and how could) they opt out that aggregated anonymised data is used?

22. Could you please explain the approach for the comparators (L423-424): Why did you censor and include them into the eczema group instead of excluding at all?

23. Could you please elaborate the ‘outcomes’ section (L435ff.)? It seems crucial to understand why you were looking this broadly for diagnosed secondary diseases, procedures etc. from HES data and those three different hierarchies. We suggest providing some more explanation.

24. The ‘non-consulters’ are only referred to in connection with p-values of HRs (L101). Could you please explain what this group is all about and the issue of exclusion? Might be useful in the methods section.

25. Rate/ratio (L457-459): Please consider replacing “rates” by hazards/risks.

26. Pipeline, number of cohorts (L467): You mention five cohorts at this stage and there are four cohorts in figure 1c, could you please explain?

Tables and Figures

27. Figure 1:

o The sum of the number of cases in the under 18 and over 18 ‘exposed’ individuals cannot exceed the number of all individuals ‘ever having eczema’ and should also be reflected in the ‘any age’ group. Could you please explain the differences?

o It might be useful to highlight the ‘final’ numbers of study participants.

28. Figure 8: Could you please try to improve the quality and layout an aim for no overlaps of dots and text?

29. Table 3: The nuances of the grey shades are not fully clear (values around 1.3 to 1.5 are sometimes white font, sometimes black and similarly sized values are highlighted differently).

(Remarks on code availability)

Reviewer #4

(Remarks to the Author)

(Remarks on code availability)

Version 1:

Reviewer comments:

Reviewer #2

(Remarks to the Author)

Thank you for addressing all the points comprehensively

(Remarks on code availability)

N/A

Reviewer #3

(Remarks to the Author)

The authors provided a thorough revision of the article. All comments were addressed appropriately. Regarding comment 3.25, however, there is still a desire for clarification. If this paragraph refers to converting hazard ratios into rates (we were not sure, if this was the case), could you please explain whether the relevant assumptions (e.g. proportional hazards assumption, censoring, etc.) were met? On the other hand, if these were simple rate estimates, why not estimating them "directly" (participants/patients without AD and the relevant outcome, e.g. Crohn's disease divided by all participants without AD, or participants/patients without AD, who developed the outcome of interest divided by time/person-years)? Or does this conversion only apply to the adjusted estimators? In this context, the question then arises as to the labelling of Suppl. Table 2 [Rate difference calculated using the estimated rate in the unexposed (rate in the exposed * (1/hazard ratio))] – difference between what exactly? When looking at your previous publication you already stated that <<... estimated adjusted rate differences [were calculated] based on the hazard ratio (as the rate in those without eczema times the inverse of the hazard ratio subtracted from the rate in those with eczema).>>

Could you please elaborate in more detail how you approached this and add it to the relevant methods section including references for the methodological approach used in these procedures?

(Remarks on code availability)

Reviewer #4

(Remarks to the Author)

(Remarks on code availability)

Version 2:

Reviewer comments:

Reviewer #3

(Remarks to the Author)

The authors have once again provided detailed answers to our questions.

Given that Table 2 is very central, it might be useful to insert repeating headers and make it clearer where the different cohorts begin/end. This increases the page count, but helps the reader significantly.

Only one question remains: Why did you calculate an HR (and HR differences) for Atopic Dermatitis (L20) in Supp 7 given that atopic eczema is the main exposure?

(Remarks on code availability)

Reviewer #4

(Remarks to the Author)

(Remarks on code availability)

REVIEWER COMMENTS

Reviewer #1 (Remarks to the Author):

COMMENT 1.1.: A well-written and very informative paper. The authors used public records in UK to test associations of atopic dermatitis with diseases across the full health spectrum. They clearly state the limitations and potential use of their findings.

RESPONSE 1.1: Thank you for reviewing our study.

Reviewer #2 (Remarks to the Author):

COMMENT 2.1: This is a comprehensive study, one of the largest and most credible yet conducted on AD comorbidities. The authors have a long history of credibility in these investigations and have contributed greatly to this field. The study builds on prior work, with important new findings on eye disease, confirmations and clarifications of the effect size of prior associations such as IBD, first identified in 2006 by Weidinger (PMID: 26253344). The data compiled here will have great utility in driving further work to understand the extensive co morbidities of Atopic Dermatitis. The methodologies employed are sound: the databases are very reliable, statistical methodologies applied are appropriate, well conducted and reported. I think this work is at the threshold for Nat Comms acceptance. If I were to be critical, the work may not have many standout truly novel headline findings.

RESPONSE 2.1: Thank you for reviewing our study.

Reviewer #3 (Remarks to the Author):

General Remarks

COMMENT 3.1. Matthewman et al. made an enormous effort to provide an analysis of cohort of people with eczema compared to matched comparators in millions of people with up to 25 years of individual follow-up. It is a very elaborate article that follows the previous study from Matthewman et al. 2024 (<https://doi.org/10.1038/s41467-024-54035-1>).

RESPONSE 3.1: Thank you for reviewing our study and for providing us with feedback. We have addressed all comments below which has improved our manuscript.

COMMENT 3.2. Perhaps it could be explained in more detail what exactly the added value is in presenting the hazard ratios in the various classification systems (ICD, phecodes, GBD)?

RESPONSE 3.2: We opted to use multiple different disease classification systems as each has different strengths and limitations. The Global Burden of Disease codes can give a higher-level overview and the entire results fit on a single page, as in Figure 6. While there is overlap between 3-character ICD-10

codes and phecodes, they cover some conditions differently. For example, phecode 930 “Allergic reaction to food” corresponds most closely to ICD-10 code T78 “Adverse effects, NEC”, and there is no 3-character ICD-10 code for food allergy. On the other hand, the ICD-10 code C84 “Mature T/NBK cell lymphomas” has no corresponding phecode.

CHANGE 3.2: In the Introduction we added “We used multiple disease classification systems (ICD-10, phecodes, Global Burden of Disease codes) as each has different strengths and limitations.”

COMMENT 3.3. In addition, it might be useful for readers to explain how the linkage between the primary care data and hospital records took place and how hospital patients were selected.

RESPONSE 3.3: We were directly provided with deidentified linked data by the data provider (CPRD). We did not perform the linkage ourselves. Linkage is undertaken by a trusted third party. Further detail is available from CPRD: https://www.cprd.com/sites/default/files/2024-11/HES_APC_Documentation_v2.9.pdf

CHANGES 3.3:

- In Methods > Study population we added “Deidentified linked data were provided by the CPRD.”
- We added the following reference: 28. Clinical Practice Research Datalink. Hospital Episode Statistics (HES) Admitted Patient Care and CPRD primary care data Documentation. https://www.cprd.com/sites/default/files/2024-11/HES_APC_Documentation_v2.9.pdf (2024).

COMMENT 3.4. We were wondering whether this analysis followed a predefined protocol/analysis plan and whether the study-reporting summary would be available (we did not discover it in the review portal).

RESPONSE 3.4: The study was approved by the CPRD under protocol #23_002665, for which the lay summary is reported here: <https://www.cprd.com/approved-studies/adverse-health-outcomes-among-people-atopic-eczema-consistent-application-cohort>. We have now completed and uploaded the reporting summary.

COMMENT 3.5. Please include study design in the title so it can be detected in systematic reviews.

RESPONSE 3.5: Done.

CHANGE 3.5: We have changed the title to “Mapping risks of hospital-recorded

health conditions in people with atopic eczema: cohort studies in English primary care and hospital data”

Abstract

COMMENT 3.6. Please include risk estimates for “large relative risk increases” for IBD and eye diseases.

RESPONSE 3.6: Done.

Change 3.6: In the abstract, we added effect estimates: “[...] for inflammatory bowel conditions (e.g., K50 Crohn disease, crude hazard ratio 1.70 [1.63-1.77]) and eye diseases (e.g., H16 Keratitis, crude hazard ratio 1.71 [1.57-1.86])”

Introduction

COMMENT 3.7. In our view, the introduction does not yet sufficiently explain the rationale for this enormous effort to process this large amount of data.

RESPONSE 3.7: We agree that an important rationale for doing this study, the ability to better contextualise findings with the aim of informing priorities in clinical practice, was previously not given in the introduction. We have updated the introduction accordingly. We have also clarified that another rationale was to assess if there may be associated conditions that had not previously been considered as associated with eczema.

CHANGE 3.7:

- **In the Introduction, first paragraph, we added “There is no internationally accepted approach to screening for, or prevention of, associated outcomes. It is possible that this lack of a consensus approach to prevention is due to clinicians not knowing which conditions to prioritise.”**
- **In the Introduction, second paragraph, we changed the second sentence to read: “[Knowledge of risks across the full health spectrum faced by people with eczema is however still limited], and there may be associated conditions that had not previously been considered as associated with eczema.”**

COMMENT 3.8. Could you please elaborate in more detail the motivation of linking data collected upon hospital admission with primary care data for the purpose of learning more about risk factors for AE (L47-49).

RESPONSE 3.8: We have made edits in the Introduction to better highlight motivations of this work. Note: our motivation was not to learn about risks factors for (someone developing) AE, but rather learn about adverse outcomes of AE.

CHANGE 3.8: In the Introduction, we edited the fourth paragraph to read: “Here, we explored associations between eczema and outcomes recorded as

part of hospital admissions. The hierarchical structure of ICD-10 allowed us to systematically learn about risks across the full health spectrum. Our aim was to create a comprehensive evidence base that facilitates comparison between outcomes, prioritisation in clinical practice, and reduces the risk of ascertainment bias.”

COMMENT 3.9. Why is ascertainment bias (L37) an issue when assessing ‘outcomes’ (as per your article this would be diagnoses or procedures in connection with secondary diseases, right?)

RESPONSE 3.9: In this sentence we refer to ascertainment bias being an issue when assessing outcomes specifically in primary care, as was done in our previous study. We elaborate on this in the second paragraph of the Discussion > Strengths and limitations section (“Ascertaining outcomes in hospital records may also make observed differences less likely to be due to ascertainment bias (e.g., people with eczema may consult their GP more frequently because of their eczema but are unlikely to be admitted to hospital more frequently”).

CHANGE 3.9: In the Introduction, second paragraph, we added “[Furthermore, ascertainment bias may be an issue when assessing outcomes in primary care] as individuals with eczema may consult more frequently due to their eczema.”

Results

COMMENT 3.10. The results section contains a lot of technical explanations, which in our view should be part of the methods section. This leaves little room for the key results, which readers have to somewhat ‘work out’ from the supplement tables and the dashboard.

RESPONSE 3.10: We agree that the results section contains technical explanations. This is due to the article structure required by the journal where the results section comes before the methods section. We have now edited to move some of the technical explanations to the methods section, but have retained a minimal set of explanations in the results sections so that readers are able to read the results section before the methods section.

CHANGES 3.10:

- In Results > Descriptive Statistics we shortened the description of the eczema definition to “(>1 record for eczema and >2 records for eczema therapies in primary care)”.
- In Results > Descriptive Statistics we removed the following explanation of censoring: “(died, left the GP practice, last data collection from practice, or the 31st March 2023)”
- In Results > Associations between eczema and subsequent hospital-recorded diagnoses, we removed details on comorbidity-variables and sensitivity analyses.

- In Methods, we added a section on “Sensitivity analyses”: “We conducted two types of sensitivity analyses. Firstly, to account for individuals already having the outcome of interest at index date, but no hospital record for it (yet) we included only individuals that had been hospitalised at least one year before index date. Secondly, to address consultation bias or differential health seeking behaviour we excluded “non-consulters”, i.e., individuals who did not have any of four common primary care records (9N11.00 - Seen in GP's surgery, 22A..00 - Body weight, 4....00 - Laboratory test, 246..00 - O/E - blood pressure reading) in the year prior to index date (as was done in our previous study).”

COMMENT 3.11. Which group is referred to by ‘same cohort’ (L163/164), the ‘any age cohort’? Therefore, comparisons with previous results could be better placed in the discussion than in the results section.

RESPONSE 3.11: In the previous study in primary care, we also created multiple cohorts in the same way as this study (i.e., any-age cohort, <18 cohort, 18+ cohort and 40+ cohort).

CHANGE 3.11:

- In Discussion > Comparison with results with primary care outcomes, we changed “in the same cohort” to “using the same cohort definitions (i.e., any-age, <18, 18+, 40+) ”.
- We have moved the section “Comparison with results with primary care outcomes” from the Results section to the Discussion section and merged it with existing text on the comparisons in the Discussion section: “We previously assessed associations with 71 outcomes recorded in primary care in the same cohorts (without requiring eligibility for HES linkage). In Figure 8 we compare our results against these previous results (for the 69 outcomes we were able to identify a corresponding ICD-10 code or phecode), using the same adjustment set. Results were generally similar (previous vs this study e.g., food allergy HR [99%CI] 4.03 [3.95-4.11] vs 930 Allergic reaction to food 4.37 [4.11-4.63]; Diverticular disease 1.17 [1.16-1.18] vs K57 Diverticular disease of intestine 1.17 [1.16-1.19]; Prostate cancer 1.01 [0.99-1.03] vs C61 Malignant neoplasm of prostate 1.04 [1.01-1.06]). Results were also generally similar for outcomes unlikely to be primary reasons for hospital admission (previous vs this study e.g., Hypertension 1.11 [1.10-1.12] vs 1.19 [1.18-1.20]), suggesting these diagnoses are recorded during hospital admissions, even if they are not the primary reason for admission.”

COMMENT 3.12. We really liked the dashboard. It remains to be clarified whether the editors will allow this as a regular part of the article.

RESPONSE 3.12: Thank you. Indeed, it remains to be clarified to what extent we are allowed to refer to the dashboard in the article. Currently we refer to

“Interactive Figures”. We believe these figures offer readers a good way of browsing and interacting with our results while referring to the manuscript.

Discussion

COMMENT 3.13. The summary of findings is well organised, but these results are not to be found in the text of the results section, but (as mentioned above) are to be searched for by the reader in various tables.

RESPONSE 3.13: We intentionally did not refer to the specific results from the summary in the results section as we were concerned this would constitute selective reporting. Given the large number of results, we have kept the results section relatively high-level, and chose to highlight what we felt to be the most important results in the Discussion section. If requested, we would be happy to include a subsection in the Results, e.g., “Most important results”.

COMMENT 3.14. It could be useful to incorporate the structure of the summary of most important findings (ophthalmologic outcomes, signals for Crohn’s disease, relatively small outcome risks comparing people with/without eczema in childhood) again for the section from L294 on.

RESPONSE 3.14: We agree and have edited accordingly.

CHANGE 3.14: in Discussion > Discussion of most important findings, we have moved the paragraph on diseases of the eye first, and the paragraphs on digestive system disorders second.

COMMENT 3.15. Strengths and limitations have been considered in a very detailed and thoughtful way.

RESPONSE 3.15: Thank you.

COMMENT 3.16. Comparison of U18-cohort with others (L250ff.): “Several factors may explain ...” it is hard to find those comparisons in the results section.

RESPONSE 3.16: While we include several example comparisons between the <18 and any-age cohort in the results sections, we are again limited in reporting the full results in the results section, as also discussed in REPONSE 3.13. However, we believe that especially for comparisons between the <18 cohort and any-age cohort Figures 2-4 offer a good visualisation of the argument in the discussion section, which we have now highlighted.

CHANGE 3.16: In Discussion > Comparisons between cohorts, we have highlighted the relevant Figures: “In the <18 cohort, the proportion of significant outcomes was lower (Figure 2), absolute risk increases were less distributed across organ systems (Figure 3), and several outcomes found to

be associated in the other cohorts were not associated in the <18 cohort (Figure 4).”

COMMENT 3.17. How realistic is it to have an even longer observation period than 25 years? The U18 cohort is already the one for which the individual observation period was probably the longest? Which secondary/concomitant diseases of AE that only occur in old age would be relevant from your point of view?

RESPONSE 3.17: The median (IQR) follow-up time is 5.2 (2.0, 10.8) in the <18 cohort and 5.5 (2.1, 11.0) in the any-age cohort, but given the large sample size there are many individuals with longer follow-up time. We agree that up to 25 years of follow-up time is a suitable timeframe to evaluate many outcomes secondary to atopic eczema. One mechanism through which outcomes may be associated with eczema is chronic systemic inflammation (e.g., see Oliveira C, Torres T. More than skin deep: the systemic nature of atopic dermatitis. Eur J Dermatol. 2019), although we acknowledge the systemic nature of eczema is not fully understood.

CHANGE 3.17: In Discussion > Interpretation of findings > Comparisons between cohorts, we added: “[Firstly, follow-up extending a maximum of 25 years may be too short to observe outcomes occurring most commonly in older age], or outcomes for which the risk may only increase after many years of living with eczema (potentially due to chronic systemic inflammation).”

COMMENT 3.18. The age at the time of the index was significantly younger (see Table 1). Did you consider, particularly in the U18 cohort, that these individuals might have had fewer hospitalisations and therefore fewer ‘outcomes’ in general?

RESPONSE 3.18: Given the age-matching calliper of 2 years, individuals within a matched set were always within 2 years of age from each other. Consequently, any substantial difference in hospitalisation rates based on age are unlikely.

COMMENT 3.19. The chain of associations that the in-hospital diagnosed anaemia (L341) is a direct consequence of nutritional deficiency caused by food allergy caused/triggered by AE is somewhat speculative from our point of view.

RESPONSE 3.19: We agree and have removed the sentence.

CHANGE 3.19: We removed the sentence “The increased risk of anaemias may be due to nutritional deficiencies related to food allergy or intolerance.”

Conclusion

COMMENT 3.20: In what way do you think your results could inform guidelines (L391)

RESPONSE 3.20: We believe one of the key advantages of our work in informing clinical guidelines (in comparison to single-outcome studies) is the ability to compare across outcomes. We have edited the conclusion section to highlight this. Examples of how our findings could inform guidelines include comorbidity screening (e.g., which comorbidity symptoms should be asked about, should any tests be performed), and multidisciplinary referral strategies (e.g., when should a patient be referred to ophthalmology, for which we mention recent guidelines for dermatologists in the Nordic region earlier in the discussion).

CHANGE 3.20: We changed the conclusion to “In conclusion, our study to our knowledge is the first to offer both a big-picture view of risks faced by people with eczema across the entire disease spectrum, as well as allowing detailed inspection of each individual association. Besides filling in gaps in knowledge, the comparable estimates facilitate prioritisation in clinical practice, including through informing guidelines (e.g., on comorbidity symptom screening or multidisciplinary referrals). Incorporating all diseases and allowing for comparisons make our findings a hypothesis-generating tool for future studies on any eczema-associated disease outcome.”

Methods

COMMENT 3.21. Anonymised/pseudonymised data (L402-404): It might be confusing to read these two sentences, particularly for readers who are not experts of NHS data. Can patients generally opt out, so that their data will not be used for research purposes? Why should (and how could) they opt out that aggregated anonymised data is used?

RESPONSE 3.21: We agree that details on the opt-out process may be more confusing than helpful in the methods section. We have therefore removed these. Manuscripts submitted to Nature journals should include a statement affirming that informed consent was obtained from all human research participants, which is why we have kept the sentence “Individual patient consent is not required or possible since CPRD provides deidentified data”. The Ethics section is now closely in line with that of our previous study, already published in Nature Communication. Patients can opt out of having their data used for research purposes, via the national opt out register, and processes for opt-out are described on the CPRD and opt-out register websites.

CHANGE 3.21: In Methods > Ethics, we removed “Consent is given by the GP practices that contribute data to CPRD. Individual patient consent is implied. However, patients are offered the right to opt out from the use of their pseudonymised data.”

COMMENT 3.22. Could you please explain the approach for the comparators (L423-424): Why did you censor and include them into the eczema group instead of excluding at all?

RESPONSE 3.22: Not allowing comparators to subsequently be included in the eczema group when they meet the eczema definition could have introduced selection bias or reduced sample size. For example, censoring and then excluding individuals that initially do not have eczema when they meet the eczema definition would reduce the size of the eczema group. If we had not made these individuals available for matching before they meet the eczema definition, this would have been using future information, thus introducing immortal time bias. We have clarified the sentence on re-matching.

CHANGE 3.22: In Methods > Study population, we changed the sentence from “Comparators were censored on the day they met the eczema definition themselves and were then included in the eczema cohort.” to “[Comparators were censored on the day they met the eczema definition themselves and] could then be re-matched as exposed individuals.”

COMMENT 3.23. Could you please elaborate the ‘outcomes’ section (L435ff.)? It seems crucial to understand why you were looking this broadly for diagnosed secondary diseases, procedures etc. from HES data and those three different hierarchies. We suggest providing some more explanation.

RESPONSE 3.23: In the Methods > Outcomes section we added an explanation of why we looked broadly across all outcomes (i.e., to learn about risks across the full health spectrum). The justification for looking broadly at outcomes across the full health spectrum is provided in the Introduction. We have also further elaborated on the justification in response to COMMENTS 3.2, 3.7, and 3.8.

CHANGE 3.23: In Methods > Outcomes, we added: “To systematically learn about risks across the full health spectrum we used multiple disease classification systems, each with different strengths and limitations:”

COMMENT 3.24. The ‘non-consulters’ are only referred to in connection with p-values of HRs (L101). Could you please explain what this group is all about and the issue of exclusion? Might be useful in the methods section.

RESPONSE 3.24: We have now clarified what “non-consulters” means in the Results section.

CHANGE 3.24: In Results > p-values, after “excluding non-consulters”, we added: “(individuals who likely did not see their GP in the year prior to index date).”

COMMENT 3.25. Rate/ratio (L457-459): Please consider replacing “rates” by hazards/risks.

RESPONSE 3.25: We believe that “rates” rather than “hazards” is the correct term here, since the rate in people with eczema was not derived from Cox

regression but rather from a simple rate estimates (i.e., the number of events per time).

COMMENT 3.26. Pipeline, number of cohorts (L467): You mention five cohorts at this stage and there are four cohorts in figure 1c, could you please explain?

RESPONSE 3.26: We considered analyses with the cohort of ever-hospitalised individuals a sensitivity analysis, which is why this cohort is not shown in Figure 1c. We have clarified this in the Pipeline section.

CHANGE 3.26: We changed mentions of “five cohorts” to “for all four cohorts (any-age, <18, 18+, 40+) and an additional cohort for sensitivity analyses (ever-hospitalised)”, and “all four cohorts and the ever-hospitalised cohort”.

Tables and Figures

COMMENT 3.27. Figure 1:

- o The sum of the number of cases in the under 18 and over 18 ‘exposed’ individuals cannot exceed the number of all individuals ‘ever having eczema’ and should also be reflected in the ‘any age’ group. Could you please explain the differences?
- o It might be useful to highlight the ‘final’ numbers of study participants.

RESPONSE 3.27: The reason for the sum of the <18 exposed and the 18+ exposed counts exceeding the any-age count is that the 18+ cohort includes incident and prevalent eczema, i.e., an individual meeting the eczema definition before their 18th birthday would be included in the <18 cohort, and would also be included in the any-age cohort from their 18th birthday. The final number of study participants varies for each outcome, as individuals who have the outcome before index date are excluded. We include examples of this in figure 1c at the bottom of the flowchart.

COMMENT 3.28. Figure 8: Could you please try to improve the quality and layout an aim for no overlaps of dots and text?

RESPONSE 3.28: Done.

CHANGE 3.28: We have fixed the overlap of dots and text, and darkened the light grey text to improve legibility.

COMMENT 3.29. Table 3: The nuances of the grey shades are not fully clear (values around 1.3 to 1.5 are sometimes white font, sometimes black and similarly sized values are highlighted differently).

RESPONSE 3.29: Only the cell background is shaded according to the value. The cell text is shaded for best legibility depending on the background. The table will likely be reformatted by the journal.

Reviewer #4 (Remarks to the Author):

COMMENT 4.1: I co-reviewed this manuscript with one of the reviewers who provided the listed reports. This is part of the Nature Communications initiative to facilitate training in peer review and to provide appropriate recognition for Early Career Researchers who co-review manuscripts.

RESPONSE 4.1: Thank you for reviewing our manuscript.

FOR REFERENCE – ORIGINAL TEXT IN LINES 457-459: Since comparators were matched, we estimated rates of outcomes among patients without eczema as the rate in people with eczema multiplied by the inverse of the corresponding hazard ratio.

FOR REFERENCE - ORIGINAL COMMENT 3.25. Rate/ratio (L457-459): Please consider replacing “rates” by hazards/risks.

FOR REFERENCE - ORIGINAL RESPONSE 3.25: We believe that “rates” rather than “hazards” is the correct term here, since the rate in people with eczema was not derived from Cox regression but rather from a simple rate estimate (i.e., the number of events per time).

REVIEWER COMMENTS

Reviewer #3 (Remarks to the Author):

The authors provided a thorough revision of the article. All comments were addressed appropriately. Regarding comment 3.25, however, there is still a desire for clarification. If this paragraph refers to converting hazard ratios into rates (we were not sure, if this was the case), could you please explain whether the relevant assumptions (e.g. proportional hazards assumption, censoring, etc.) were met? On the other hand, if these were simple rate estimates, why not estimating them “directly” (participants/patients without AD and the relevant outcome, e.g. Crohn’s disease divided by all participants without AD, or participants/patients without AD, who developed the outcome of interest divided by time/person-years)? Or does this conversion only apply to the adjusted estimators? In this context, the question then arises as to the labelling of Suppl. Table 2 [Rate difference calculated using the estimated rate in the unexposed (rate in the exposed * (1/hazard ratio))] – difference between what exactly? When looking at your previous publication you already stated that <<... estimated adjusted rate differences [were calculated] based on the hazard ratio (as the rate in those without eczema times the inverse of the hazard ratio subtracted from the rate in those with eczema).>>

Could you please elaborate in more detail how you approached this and add it to the relevant methods section including references for the methodological approach used in these procedures?

RESPONSE: Thank you for your thorough review which has helped us improve the manuscript. We are pleased that we were able to address your comments. Regarding your original comment 3.25, we now provide additional clarification here.

The rate in the exposed and the (observed) rate in the unexposed are indeed simple rate estimates and are both estimated directly (number of people who developed the outcome per person-years). From this we also calculate the observed rate difference by subtracting one from the other.

The reason to estimate rates in the unexposed as the rate in the exposed multiplied by the inverse hazard ratio is that unexposed individuals are matched to exposed individuals and we wish to draw comparisons within these matched sets. The simple rate in the unexposed ignores the matched sets. However, the hazard ratio is estimated from Cox regression stratified on matched set. Therefore, the rate difference estimated using the hazard ratio can be seen as approximating a comparison within matched sets, while the observed rate difference cannot.

An illustrative example of how this can make a difference is outcome F83 (Mixed specific developmental disorders), where we estimated a crude hazard ratio of 0.95 (i.e., the hazard was lower in people with eczema). However, the observed rate difference is 0.00184, indicating an excess rate in people with eczema, which contradicts the hazard ratio. The estimated rate difference is -0.00186 which is in line with the hazard ratio. (We also re-estimated the hazard ratio omitting the stratification (+strata(setid) in the R survival package) which resulted in a hazard ratio of 1.16, confirming that the sign reversal in this example was due to not accounting for matching.)

However, the differences are so small that they are rounded to 0 in Supplementary Table 2, and in general there is no difference in interpretation to our study whether we use the observed or estimated rate difference. We found only one other outcome (F16) where this mismatch in sign between observed and estimated rate difference occurs.

For the reasons above, we prefer to keep using the estimated rate in the unexposed. In addition, this approach allows us to estimate rate differences using both the crude and the adjusted hazard ratio. For transparency we report observed and estimated rate differences.

Note: In general, since this is a matched cohort study, any rates in the unexposed population should not be interpreted as rates in the general population. To clarify this, we have changed the sentence in lines 457-459.

We agree that the labelling in Supplementary Table 2 could be improved and have made changes accordingly.

CHANGES:

- Manuscript: We changed "*Since comparators were matched, we estimated rates of outcomes among patients without eczema as the rate in people with eczema multiplied by the inverse of the corresponding hazard ratio.*" to "*Since comparators were matched, we estimated rates of outcomes expected in the absence of eczema as the rate in people with eczema multiplied by the inverse of the corresponding hazard ratio.*"
- Supplementary Table 2: We have renamed the "Crude Rate (per 1,000 person-years)" spanner to "Rate (per 1,000 person-years)"
- Supplementary Table 2: We changed the word "minimally-adjusted" to "crude", as we do not use the word "minimally-adjusted" elsewhere in the manuscript, and we already explain that crude results are "*implicitly*

adjusted through matching on age, sex and general practice, and calendar time, as comparators entered the cohort on the same day as exposed individuals” in the Statistical analysis section of the manuscript.

- **Supplementary Table 2: We have added the following footnotes explaining how each of the columns is derived:**
 - **Hazard ratio (99% confidence interval): Hazard ratios (99% confidence intervals) estimated from Cox models (stratifying on matched set) comparing people with eczema to those without eczema. Results are implicitly adjusted through matching on age, sex and general practice, and calendar time, as comparators entered the cohort on the same day as exposed individuals. * indicates that the result is significant under Bonferroni-correction.**
 - **Crude rate in the unexposed (estimated): Estimated as the rate in the exposed * (1/crude hazard ratio)**
 - **Rate difference (observed): Rate in the exposed - observed rate in the unexposed**
 - **Rate difference (crude estimated): Rate in the exposed - rate in the exposed * (1/crude hazard ratio)**
 - **Rate difference (adjusted estimated): Rate in the exposed - rate in the exposed * (1/comorbidity-adjusted hazard ratio)**

Reviewer #3 (Remarks to the Author):

The authors have once again provided detailed answers to our questions.

Given that Table 2 is very central, it might be useful to insert repeating headers and make it clearer where the different cohorts begin/end. This increases the page count, but helps the reader significantly.

Only one question remains: Why did you calculate an HR (and HR differences) for Atopic Dermatitis (L20) in Supp 7 given that atopic eczema is the main exposure?

RESPONSE: We systematically calculated and reported results for all ICD-10 codes, including L20. For Table 1 (previously Table 2) we have added a description of where the different cohorts begin and end.